# SpecEM: Training-Free LLM Ensembling via Iterative Drafting, Verification, and Online Feedback

**Bo Lv**[1,2,3], **Nayu Liu**[4*], **Chen Tang**[5], **Xin Liu**[1], **Yue Yu**[1*], **Ping Luo**[1,2,3]

[1]Peng Cheng Laboratory
[2]Key Lab of Intelligent Information Processing, Institute of
Computing Technology, Chinese Academy of Sciences (ICT/CAS)
[3]University of Chinese Academy of Sciences
[4]Tianjin Laboratory Autonomous Intelligence Technology and Systems, School of
Computer Science and Technology, Tiangong University
[5]Institute for Advanced Algorithms Research, Shanghai
`lvbo19@mails.ucas.ac.cn`

## Abstract

Ensembles of generative large language models (LLMs) are a promising way to compensate for individual model limitations, integrating the strengths of different LLMs. Existing LLM ensemble methods, however, face limitations such as first-token delay and challenges in long-range semantic collaboration between models, Moreover, they typically assume equal voting weights for all models during ensemble, ignoring task-specific performance differences among models. In this work, we propose SpecEM, a training-free, plug-and-play LLM ensemble framework that dynamically adjusts each model's model contribution in real time based on task performance. Inspired by speculative decoding, SpecEM iteratively performs drafting and verification, allowing models to collaborate semantically at the segment level for integrated output. Furthermore, we introduce an online feedback mechanism with multiplicative weight updates, where each model's voting weight is adjusted on-the-fly according to how often it outperforms others during verification stage, ensuring that stronger models exert greater influence during ensembling. Experimental results on five LLM families (ranging from 7B to 72B parameters) and six benchmark datasets, spanning open-domain instruction following, reasoning, commonsense, demonstrate consistent performance improvements compared to state-of-the-art LLM ensemble methods. Our code is available at https://github.com/lvbotenbest/SpecEM.

## 1 Introduction

Generative large language models (LLMs) [AI@Meta, 2024, Yang et al., 2024] have been widely adopted due to their impressive performance across a broad range of domains. Owing to differences in training data and model architectures, off-the-shelf generative LLMs often exhibit strengths in different areas. Consequently, ensembling multiple LLMs at inference time can help mitigate individual biases and errors, resulting in a more robust and reliable user experience.

Existing LLM ensemble approaches [Chen et al., 2025, Jiang et al., 2023b, Huang et al., 2024, Lv et al., 2024a] can be broadly categorized into generate-then-ensemble [Jiang et al., 2023b, Lv et al., 2024b] and ensemble-while-generation [Huang et al., 2024, Yao et al., 2025] paradigms. The former typically generates full responses from all base models for a given query, then leverages an additional

---

*Corresponding author.

39th Conference on Neural Information Processing Systems (NeurIPS 2025).

fusion model to summarize or select the best outputs. The latter adopts a more interleaved approach, greedily aggregating the output probabilities from different models at certain timesteps to decide next tokens, which is then broadcast to all models.

Despite promising progress, generate-then-ensemble methods suffer from first-token delay, as users must wait until all models complete their responses before receiving integrated output. In contrast, ensemble-while-generation methods mitigate this latency but may fall short in enabling long-range semantic communications across models. Moreover, existing methods focus solely on aggregating model outputs, typically assuming equal contribution from all models, and overlooking the fact that different models may perform differently depending on the task. We argue that incorporating an online learning mechanism to dynamically assign higher weights to better-performing models while down-weighting weaker ones can lead to higher-quality ensemble outputs.

Based on the above observations, we propose SpecEM, a training-free, plug-and-play LLM ensemble framework that performs segment-level fusion of model outputs and dynamically adjusts model weights on-the-fly based on task-specific performance. Inspired by speculative decoding [Xia et al., 2023, Leviathan et al., 2023a], SpecEM iteratively executes two key stages: drafting and verification. In the drafting stage, each base LLM generates a candidate text segment given the prior context, with a predefined maximum length per iteration. In the verification stage, all LLMs receive these candidate segments with prior context and mutually evaluate them in parallel based on their output logits. The top-ranked segment is then broadcast to all models, guiding them to generate higher-quality text in subsequent rounds. This iterative drafting-verification process eliminates the need for training fusion modules or selection aggregators, and allows for effortless integration of base LLMs without any fine-tuning.

Furthermore, we introduce an online feedback mechanism in SpecEM to dynamically adjust each model's influence during the verification stage. It is based on the assumption that models capable of generating higher-quality segments in drafting are also better at evaluating others in verification. Specifically, we treat the number of times a model outperforms its peers in verification as a reward signal to update its voting weight using a multiplicative weight update algorithm. This ensures that stronger models progressively exert greater influence during generation.

We evaluate SpecEM on five popular LLM families (ranging from 7B to 72B parameters) across six benchmark datasets, covering open-domain instruction following, reasoning, and commonsense. Experimental results demonstrate consistent performance improvements over state-of-the-art LLM ensemble methods. In summary, our contributions are as follows:

- We propose SpecEM, a training-free and plug-and-play ensemble framework that integrates outputs by iteratively coordinating drafting and verification across multiple LLMs.

- We propose an online feedback mechanism that dynamically adjusts each model's contribution to inference and verification during generation, ensuring that stronger models exert greater influence in the ensemble.

- We conduct comprehensive evaluations on five LLM families and six benchmarks, showing that SpecEM consistently outperforms state-of-the-art ensemble methods.

## 2 Related Work

### 2.1 LLM Ensembling

Recent efforts in LLM ensembling have shown that combining multiple models can enhance performance by leveraging their complementary strengths. These methods can be broadly categorized into generate-then-ensemble and ensemble-while-generation, depending on when and how the ensembling of model outputs occurs.

Generate-then-ensemble methods first let each base LLM generate a complete response, and then aggregate the outputs through selection [Freitag et al., 2023] or fusion [Lv et al., 2024b]. Selection-based methods, such as MBR [Freitag et al., 2023] and PairRank [Jiang et al., 2023b], rank or compare candidates among all outputs to select the best one as the output. Fusion-based methods, such as GenFuse [Jiang et al., 2023b] and MOA [Wang et al., 2024], generate new outputs by using base model responses as input to a fusion model or aggregator.

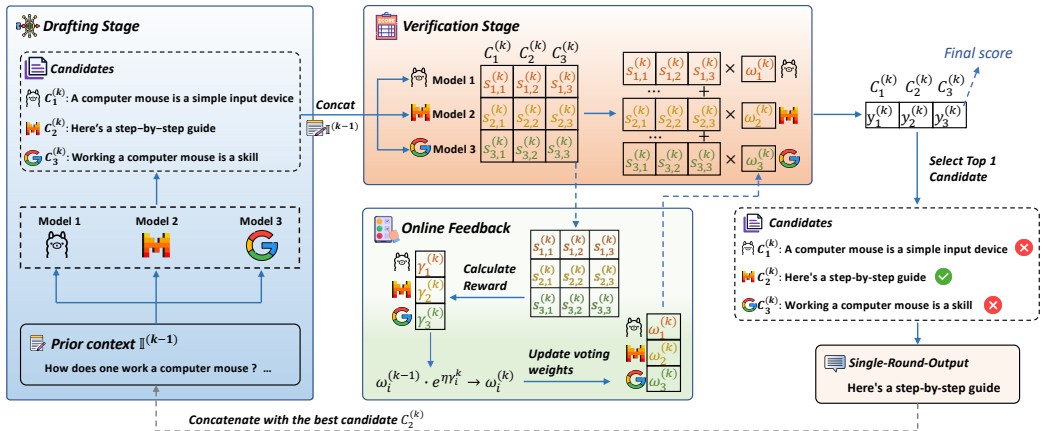

Figure 1: Overview of SpecEM, a training-free plug-and-play LLM ensemble framework with three components: drafting, verification, and online feedback. The blue solid lines indicate a single iteration; dashed lines denote input refreshing for the next round.

Ensemble-while-generation methods [Yu et al., 2024, Huang et al., 2024, Xu et al., 2024] aggregate model outputs during the generation process, typically by fusing output probability distributions to produce ensembling results incrementally. Due to vocabulary mismatches across different LLMs that hinder the combination of multiple probability distributions, Yu et al. [2024] construct a new union vocabulary by combining the vocabularies of multiple models to include all tokens from each model. They then project the distribution information from each model onto this merged vocabulary for averaging aggregation. Similarly, DeePEn [Huang et al., 2024] and EVA [Xu et al., 2024] project the output distributions of multiple models into a shared relative/pivot space, followed by averaging aggregation. While, these methods operate over all LLM's vocabulary at each timestep, which may leads to some computational overhead. More efficiently, UniTe [Yao et al., 2025] focuses only on the top-K portion of each model's output distribution and uses the union vocabulary strategy [Yao et al., 2025] to reduce alignment costs while maintaining good performance.

While recent progress has been made, challenges remain in balancing efficiency with effective cross-model collaboration. In this work, we introduce SpecEM, a plug-and-play training-free framework that performs segment-level collaboration via iterative drafting and verification. Unlike previous methods that rely on static model contributions, SpecEM incorporates an online feedback mechanism to dynamically adjust each model's influence based on performance during generation, promoting more adaptive and effective ensembling.

## 2.2 Speculative Decoding

Speculative Decoding [Xia et al., 2023, Chen et al., 2023, Sun et al., 2024] aims to accelerate inference in LLMs by leveraging a lightweight draft model to propose multiple candidate tokens, which are then verified by a larger target model [Leviathan et al., 2023b]. Concretely, at each decoding step, the draft model efficiently generates a sequence of potential tokens, and the target model accepts only those that match its own predictions [Miao et al., 2024]. This significantly reduces the number of expensive forward passes through the larger model without compromising output quality.

Inspired by this idea, we propose SpecEM, which reimagines speculative decoding for model ensembling rather than acceleration. Instead of a small model drafting for a large one, multiple LLMs iteratively generate draft segments and verify each other's outputs in parallel. This collaborative refinement allows stronger models to guide weaker ones. In addition, SpecEM incorporates an online feedback mechanism that dynamically adjusts each model's influence during verification.

## 3 Methodology

Figure 1 presents an overview of SpecEM. SpecEM performs LLM ensembling through iterative drafting, verification, and online feedback, which are described in detail in the following subsections.

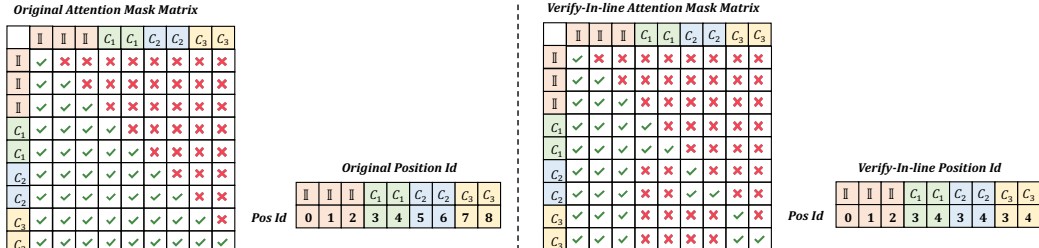

Figure 2: An overview of verify-in-line attention mask and postion id.

## 3.1 Drafting Stage

During the drafting stage, in each generation round, all models are simultaneously activated to perform parallel inference based on the task query and the best candidate segment broadcast from previous rounds. Formally, let the ensemble consist of models $\{M_i\}_N$, where $M_i$ denotes the $i$-th base LLM. In the $k$-th iteration round, the draft candidate segment generated by $M_i$ is denoted as $C_i^{(k)} = \{t_1^{(k)}, ..., t_l^{(k)}\}$, where the number of generated tokens $l$ is constrained by a predefined maximum segment length $L$.

$$C_i^{(k)} = M_i(I^{(0)}, ..., I^{(k-1)}) \tag{1}$$

Here, $I^{(k-1)}$ denotes the best candidate segment broadcast in the $k$-th round, and $I^{(0)}$ corresponds to the initial task query.

## 3.2 Verification Stage

During the verification stage, all models perform mutual evaluation on the candidate segments generated during the drafting stage. At iteration $k$, each model receives the full set of candidate segments $\{C_i^{(k)}\}_N$, along with the prior context $\{I^{(0)}, ..., I^{(k-1)}\}$. Each model scores the candidates, and the one with the highest aggregated score is selected as the output of the current round. This selected segment is then broadcast to all models as part of the context for the next generation round. As illustrated in the mutual evaluation matrix in Figure 1, let $s_{i,j}^{(k)}$ denote the score assigned by model $i$ to candidate $j$ in generation round $k$. Concretely, for each candidate segment, we compute the average of the logits produced by the model over the tokens in the segment, which serves as the model's evaluation score [Lv et al., 2023, Varshney et al., 2023, Lv et al., 2025] for that candidate:

$$s_{i,j}^{(k)} = \frac{1}{l} \sum_{u=1}^{l} p(t_{i,j,u}^{(k)}) \tag{2}$$

where $p(t_{i,j,u}^{(k)})$ denotes the logit score by model $i$ in round $k$ for the $u$-th token of candidate $j$. To mitigate scoring bias caused by some models producing systematically higher or lower logits, we normalize the scores that the model assigns to all candidates before aggregation:

$$s_{i,j}^{(k)} \leftarrow \frac{s_{i,j}^{(k)}}{\sum_{j=1}^{N} s_{i,j}^{(k)}} \tag{3}$$

Finally, the overall score $\{y_j\}_N$ of each candidate is computed as a weighted sum of the scores from all verifier models:

$$y_j^{(k)} = \sum_{i=1}^{N} \omega_i^{(k)} s_{i,j}^{(k)} \tag{4}$$

with weights $\{\omega_i\}_N$ dynamically updated by the online feedback mechanism described in Section 3.3.

Specially, we introduce a verify-in-line mechanism, avoiding redundant attention computations over the prior context and the increased time complexity caused by serial model-wise scoring. Concretely, at the $k$-th generation iteration, we concatenate the prior context and all candidate segments along the sequence dimension to construct a unified input sequence:

$$LINE = [\mathbb{I}^{(k-1)} : C_1^{(k)} : C_2^{(k)} : C_N^{(k)}] \tag{5}$$

where $\mathbb{I}^{(k-1)}$ denotes the prior context $[I^{(0)} : ... : I^{(k-1)}]$ for brevity, and $[:]$ denotes sequence-wise concatenation. We then modify the attention mask and position IDs in the Transformer such that each model can process LINE and output scores for all candidates in parallel, as shown in Figure 2:

**Verify-in-line attention mask.** During verification, each candidate segment should only attend to the shared prior context $\mathbb{I}^{(k-1)}$, without accessing information from other candidate segments. To enforce this, we augment the standard triangular attention mask of the decoder-only Transformer with an additional masking scheme that blocks attention across candidate segments. This ensures that tokens in $C_i^{(k)}$ can only attend to $\mathbb{I}^{(k-1)}$, not to tokens in other candidates $C_j^{(k)}$ for $j \neq i$. As a result, although all candidates are concatenated into LINE, model $M_i$ effectively "sees" only $[\mathbb{I}^{(k-1)} : C_i^{(k)}]$ during scoring, enabling efficient and parallel evaluation.

**Verify-in-line position IDs.** While the modified attention mask guarantees the correct visibility, the default position encoding would still reflect the segments' physical positions in the concatenated LINE, which may distort modeling. For instance, the actual input sequence for scoring $C_2^{(k)}$ is $[\mathbb{I}^{(k-1)} : [mask] : C_2^{(k)} : [mask] : ...]$ rather than $[\mathbb{I}^{(k-1)} : C_2^{(k)}]$. To address this, we further adapt LLMs' relative positional encoding so that each candidate segment is positioned as if it were immediately following the prior context. That is, the position IDs for each $C_i^{(k)}$ are reset to be consecutive with $\mathbb{I}^{(k-1)}$, ensuring the position modeling remains consistent with the actual evaluation context.

### 3.3 Online Feedback Mechanism

It is generally difficult to anticipate which model performs best on a given query. Due to differences in model architectures and training corpora, different models may exhibit varying strengths across domains, and may produce low-quality outputs when encountering unfamiliar or challenging topics. Since the verification stage relies on each model's scoring of candidate segments, models with weaker generation capabilities may also produce unreliable evaluations when acting as validators.

We propose a core assumption: **models that perform better in generation tend to offer more reliable judgments during verification.** This assumption is empirically supported in **Appendix** A. Building on this, we introduce an online feedback mechanism based on the multiplicative weights update algorithm. It dynamically adjusts each model's contribution in the verification stage by tracking and weighting its validation performance during the generation process. As a result, decisions from better-performing models are prioritized, enhancing the overall ensemble effectiveness.

Formally, let there be $N$ models. Denote the verification weight of model $M_i$ at generation round $k$ as $\omega_i^{(k)}$. All models are initially assigned uniform weights: $\omega_i^{(0)} = \frac{1}{N}$. At round $k$, model $M_i$ receives a feedback reward $\gamma_i^{(k)}$, and its weight is updated according to:

$$\omega_i^{(k)} = \omega_i^{(k-1)} \cdot e^{\eta \gamma_i^{(k)}} \tag{6}$$

To address the fact that more models (i.e. $N$) lead to smaller initial weights, and to ensure that updates become more stable over time (i.e. $k$), we define the learning rate $\eta$ in Eq. 6 as:

$$\eta = \alpha \cdot \frac{\sqrt{1/k}}{N} \tag{7}$$

where $\alpha$ is a hyperparameter. Then all weights are normalized as:

$$\omega_i^{(k)} \leftarrow \frac{\omega_i^{(k)}}{\sum_{j=1}^{n} \omega_j^{(k)}} \tag{8}$$

This feedback-driven reweighting ensures that more credible validators exert greater influence on the selection process, leading to progressively improved collective decisions over time.

**Reward Definition.** For each model $M_i$, we define its reward $\gamma_i^{(k)}$ in Eq. 6 based on how often its generated segment is preferred over others in the evaluations conducted by the remaining models. Specifically, we count the number of times model $M_i$'s candidate $C_i^{(k)}$ is scored higher than another

candidate $C_r^{(k)}$ by a third model $M_j$, where $j \neq i$ and $r \neq i$, and normalize this count as reward $\gamma_i^{(k)}$:

$$\gamma_i^{(k)} = \frac{\sum_{j \neq i} \sum_{r \neq i} bool(s_{j,i}^{(k)} > s_{j,r}^{(k)})}{\sum_{i=1}^{N} \sum_{j \neq i} \sum_{r \neq i} bool(s_{j,i}^{(k)} > s_{j,r}^{(k)})} \tag{9}$$

Here, $bool(\cdot)$ returns 1 if the condition inside is true, and 0 otherwise. Intuitively, if a model's candidate frequently outperforms others in peer evaluations, it is considered better suited to the current task. In Section 4.3, we empirically compare this reward formulation against alternative segment selection strategies. Finally, the comprehensive score for each candidate $\{y_j^{(k)}\}_N$ is computed as the weighted sum of its scores assigned by all validators, following Equation 4. The highest-scoring candidate is selected as the best output $I^{(k)}$, and it is appended to the prior context for use in the next round of drafting and verification.

# 4 Experiments

## 4.1 Experimental Setup

**Datasets.** We evaluate SpecEM on six datasets that reflect key capabilities of LLMs, including open-domain instruction following, commonsense, and reasoning. **FuseEval**: A multilingual instruction response benchmark we construct by combining Dolly-15k [Conover et al., 2023] and Alpaca-GPT4 [Peng et al., 2023] for English, and Human-Value and Ruozb from COIG-CQIA [Bai et al., 2024] for Chinese. **IFEval** [Zhou et al., 2023]: Evaluates instruction adherence under four granular settings, prompt-strict/loose, instruction-strict/loose. **AlpacaEval 2.0** [Dubois et al., 2024]: Measures alignment with human preferences via GPT-4 based pairwise comparisons against GPT-4 outputs. **MMLU** (5-shot) [Hendrycks et al., 2021] and **ARC-C** (5-shot) [Clark et al., 2018]: Multiple-choice benchmarks that test factual knowledge and general commonsense. **GSM8K** (3-shot) [Cobbe et al., 2021]: Focuses on arithmetic and multi-step reasoning through grade-school math problems. x-shot refers to providing x examples as in-context during inference. Please refer to **Appendix** B.1 for a detailed description of datasets.

**Base LLMs.** We use top-performing open-source instruction-tuned models (7B–9B) as base LLMs in our ensemble, including **Llama-3-8B-instruct** [AI@Meta, 2024], **Mistral-7B-v0.3-instruct** [Jiang et al., 2023a], **Qwen2-7B-instruct** [Yang et al., 2024], **Glm-4-9b-instruct** [GLM et al., 2024], and **Gemma-2-9b-instruct** [Gemma et al., 2024]. Moreover, to assess scalability, we also evaluate SpecEM with larger base models, including **Qwen2-72B-instruct**, **Llama3-70B-instruct**, **Qwen2.5-32B-instruct**[Qwen et al., 2025], and **Mistral-24B-instruct-2501** [Team., 2025].

**Metrics.** We follow standard evaluation protocols for each task in previous works. For FuseEval, we use BARTScore[Yuan et al., 2021], BERTScore[Zhang et al., 2019], BLEU[Papineni et al., 2002], ROUGE [Lin, 2004], and GPT4-Rank [OpenAI et al., 2024] to assess reference-based generation quality. MMLU and ARC-C report accuracy by selecting the option with the highest likelihood. For GSM8K, we compute exact match accuracy based on the predicted answer. For IFEval, we use the provided evaluation files to test under prompt/instruction-strict and -loose conditions. AlpacaEval 2.0 reports length-controlled (LC) win rates against GPT-4 outputs using `gpt-4-1106-preview`.

**Comparative methods.** We compare our proposed SpecEM with several strong recent LLM ensemble methods, including PairRank [Jiang et al., 2023b], Minimum Bayes Risk (MBR) [Freitag et al., 2023], Generation Fusion (GF) [Jiang et al., 2023b], Mixture-of-Agents (MOA) [Wang et al., 2024], Majority Voting [Davani et al., 2022], and Unite [Yao et al., 2025]. Please refer to **Appendix** B.3 for a detailed description of these baseline methods.

**Implement details.** SpecEM requires no training and operates purely during inference. All models are loaded using bfloat16 precision, with $do\_sample = True$, $temperature = 0.6$, and $top\_p = 0.9$ generation settings. For experiments with 7B–9B models, we use A100 GPUs, while larger models (24B–72B) are evaluated on H200 GPUs. The maximum number of candidate segments is set to $L = 10$, and the online feedback hyperparameter is set to $\alpha = 1$. All reported results are averaged over three independent runs to ensure stability.

Table 1: Results on the English and Chinese subsets of the FuseEval benchmark. `Pink` highlights the best overall result, and `Blue` marks the best result among base LLMs. The upward arrow ↑ means higher is better, and the downward arrow ↓ means lower is better.

| Model | ROUGE-1↑ | ROUGE-2↑ | ROUGE-L↑ | BLEU↑ | BARTScore↑ | BERTScore↑ | GPT4-Rank↓ |
|---|---|---|---|---|---|---|---|
| *English Scenario* | | | | | | | |
| **Base LLMs** | | | | | | | |
| Llama-3-8B-instruct [AI@Meta, 2024] | 25.16 | 9.77 | 23.31 | 3.57 | -2.98 | 69.99 | 9.52 |
| Glm-4-9B-instruct [GLM et al., 2024] | 25.85 | 10.26 | 23.90 | 3.48 | -2.96 | 70.51 | 9.24 |
| Qwen2-7B-instruct [Yang et al., 2024] | 26.62 | 10.81 | 24.49 | 3.86 | -2.94 | 71.44 | 8.48 |
| Gemma-2-9B-instruct [Gemma et al., 2024] | 25.31 | 10.01 | 23.59 | 4.19 | -2.93 | 71.52 | 9.01 |
| Mistral-7B-instruct-v0.3 [Jiang et al., 2023a] | 27.75 | 10.75 | 25.57 | 4.82 | -2.94 | 71.88 | 7.62 |
| **Larger LLMs** | | | | | | | |
| Llama-3-70B-instruct [AI@Meta, 2024] | 26.77 | 10.87 | 24.56 | 4.10 | -2.84 | 70.98 | 5.22 |
| Qwen2-72B-instruct [Yang et al., 2024] | 27.26 | 11.23 | 25.11 | 4.29 | -2.76 | 71.73 | 4.21 |
| Mixtral-8x7B-instruct [Jiang et al., 2023a] | 29.04 | 12.25 | 26.75 | 4.08 | -2.81 | 72.19 | 4.16 |
| **Methods of Ensembling Base LLMs** | | | | | | | |
| GF (Qwen2) [Jiang et al., 2023b] | 23.08 | 8.92 | 21.28 | 3.19 | -2.95 | 69.70 | 10.10 |
| GF (Gemma-2) [Jiang et al., 2023b] | 21.81 | 7.66 | 20.08 | 3.00 | -3.02 | 68.20 | 10.11 |
| GF (Mistral) [Jiang et al., 2023b] | 24.92 | 9.58 | 22.97 | 3.92 | -2.93 | 70.38 | 8.56 |
| MBR [Freitag et al., 2023] | 27.12 | 10.40 | 25.33 | 4.56 | -2.89 | 71.63 | 7.66 |
| PairRank [Jiang et al., 2023b] | 28.21 | 10.86 | 25.94 | 4.99 | -2.86 | 72.09 | 6.84 |
| MOA [Wang et al., 2024] | 27.61 | 11.30 | 25.47 | 5.12 | -2.88 | 71.90 | 7.48 |
| UniTE [Yao et al., 2025] | 27.81 | 10.91 | 25.73 | 4.53 | -2.90 | 71.77 | 6.00 |
| **SpecEM (Ours)** | 31.19 | 14.40 | 28.86 | 5.81 | -2.88 | 73.34 | 3.98 |
| *Chinese Scenario* | | | | | | | |
| **Base LLMs** | | | | | | | |
| Gemma-2-9B-instruct [Gemma et al., 2024] | 29.15 | 7.65 | 18.35 | 3.36 | -4.28 | 68.73 | 8.58 |
| Mistral-7B-instruct-v0.3 [Jiang et al., 2023a] | 30.99 | 8.65 | 20.66 | 4.42 | -4.48 | 70.10 | 6.55 |
| Qwen2-7B-instruct [Yang et al., 2024] | 29.93 | 8.09 | 20.03 | 3.62 | -4.33 | 69.99 | 6.42 |
| Glm-4-9B-instruct [GLM et al., 2024] | 30.88 | 8.71 | 20.42 | 4.47 | -4.30 | 70.25 | 5.18 |
| **Larger LLMs** | | | | | | | |
| Llama-3-70B-instruct [AI@Meta, 2024] | 27.78 | 7.05 | 20.22 | 4.14 | -4.55 | 68.52 | 7.38 |
| Qwen2-72B-instruct [Yang et al., 2024] | 31.44 | 8.97 | 22.48 | 4.88 | -4.34 | 70.65 | 3.63 |
| **Methods of Ensembling Base LLMs** | | | | | | | |
| GF (Mistral) [Jiang et al., 2023b] | 30.29 | 8.12 | 20.33 | 3.88 | -4.54 | 70.04 | 7.28 |
| GF (Qwen2) [Jiang et al., 2023b] | 28.69 | 7.87 | 18.93 | 3.32 | -4.41 | 69.81 | 8.41 |
| GF (Glm-4) [Jiang et al., 2023b] | 30.26 | 8.70 | 20.51 | 4.27 | -4.33 | 70.23 | 5.40 |
| MBR [Freitag et al., 2023] | 30.93 | 8.71 | 20.63 | 4.31 | -4.31 | 70.23 | 5.33 |
| UniTE [Yao et al., 2025] | 27.46 | 8.22 | 20.54 | 2.83 | -4.60 | 67.58 | 9.37 |
| MOA [Wang et al., 2024] | 30.96 | 8.50 | 20.60 | 4.36 | -4.31 | 70.11 | 5.93 |
| **SpecEM (Ours)** | 32.15 | 9.94 | 24.00 | 4.75 | -4.29 | 71.03 | 3.25 |

## 4.2 Main Results

**Results on Diverse Evaluation Benchmarks.** Table 1 presents results on the English and Chinese subsets of the FuseEval benchmark. SpecEM, built on 7B–9B base models, outperforms all individual LLMs and existing ensemble methods across all metrics.

Notably, it achieves over 3-point average gains in ROUGE-1/2/L and ranks highest on GPT4-Rank. Despite using only several 7B–9B LLMs, SpecEM performs comparably to 70B-scale single models while remaining more parameter efficient. The improvements are consistent in the English and Chinese scenarios, indicating the generalizability of SpecEM across languages. We further assess SpecEM on MMLU, ARC-C, GSM8K, and IFEval benchmarks using base

Table 3: Performance on FuseEval and AlpacaEval 2.0 benchmarks. Win rates on FuseEval are measured relative to the outputs of Qwen2-72b-instruct.

| Model | English FuseEval (winrate) | Chinese FuseEval (winrate) | AlpacaEval 2.0 (LC-winrate) | Avg |
|---|---|---|---|---|
| *Base LLMs* | | | | |
| Qwen2-72b-instruct | – | – | 38.10 | – |
| Qwen2.5-32b-instruct | 49.31 | 37.48 | 43.82 | 43.54 |
| Llama3-70b-instruct | 43.10 | 10.66 | 34.40 | 29.39 |
| Mistral-24b-instruct-2501 | 52.55 | 31.79 | 48.46 | 44.27 |
| *Methods of Ensembling Base LLMs* | | | | |
| MOA | 53.63 (+1.08) | 53.12 (+15.64) | 46.98 (-1.48) | 51.24 |
| GenFuse | 51.06 (-1.49) | 52.41 (+14.93) | 49.06 (+0.60) | 50.84 |
| UniTE | 54.79 (+2.24) | 19.13 (-18.35) | 49.20 (+0.74) | 41.04 |
| SpecEM | 55.46 (+2.91) | 56.77 (+19.29) | 51.32 (+2.86) | 54.52 |

Table 2: Results on MMLU, ARC-C, GSM8K, and IFEval benchmarks. Values in parentheses indicate performance difference from the best-performing base model in ensemble in each column.

| Model | MMLU | ARC-C | GSM8K | IFEval | |
|---|---|---|---|---|---|
| | | | | prompt-avg | instruct-avg |
| *Base LLMs* | | | | | |
| Qwen2-7B-instruct | 68.23 | 84.73 | 74.22 | 41.70 | 53.88 |
| GLM-4-9B-instruct | 67.16 | 85.15 | 71.80 | 56.01 | 67.14 |
| Gemma-2-9B-instruct | 71.51 | 88.14 | 77.26 | 61.64 | 72.26 |
| *Methods of Ensembling Base LLMs* | | | | | |
| Majority-Voting | 71.78 (+0.27) | 88.38 (+0.24) | 77.29 (+0.03) | – | – |
| MBR | – | – | 76.98 (-0.28) | 54.96 (-6.68) | 66.21 (-6.05) |
| MOA | 70.43 (-1.08) | 88.28 (+0.14) | 77.30 (+0.04) | 60.80 (-0.84) | 68.81 (-3.45) |
| UniTE | 71.94 (+0.43) | 88.54 (+0.40) | 76.52 (-0.74) | 56.72 (-4.92) | 62.08 (-10.18) |
| SpecEM (Qwen2+GLM4) | 70.73 (+2.50) | 87.54 (+2.39) | 75.44 (+1.22) | 51.15 (-4.86) | 63.07 (-4.07) |
| SpecEM (Qwen2+Gemma2) | 72.18 (+0.67) | 88.40 (+0.26) | 78.70 (+1.44) | 56.01 (-5.63) | 67.39 (-4.87) |
| SpecEM (GLM4+Gemma2) | 71.82 (+0.31) | 88.74 (+0.60) | 75.82 (-1.44) | 66.89 (+5.25) | 75.52 (+3.26) |
| SpecEM (All) | 73.01 (+1.50) | 89.08 (+0.94) | 77.41 (+0.15) | 62.11 (+0.47) | 71.56 (-0.70) |

LLMs with varied strengths.

As shown in Table 2, SpecEM consistently surpasses all baseline ensemble methods across these benchmarks. In particular, SpecEM (Qwen2 + GLM4) achieves +2.5 and +2.4 improvements on MMLU and ARC-C, respectively, leveraging complementary model capabilities. These results demonstrate the effectiveness of SpecEM across diverse task formats beyond open-ended generation.

**Scaling to Larger Models.** To further evaluate the scalability of our framework, we conduct experiments by integrating four larger LLMs ranging from 24B to 72B parameters on FuseEval and AlpacaEval 2.0. As shown in Table 3, SpecEM consistently outperforms all base models and prior ensemble baselines, achieve the best performance. In particular, it surpasses the strongest single base model by an average win rate margin of 10.3 points. These results demonstrate that SpecEM generalizes robustly across model sizes scale.

Table 4: Win-rate comparisons between SpecEM and ablations on English (EN) and Chinese (CN) FuseEval. Each cell shows the win/loss percentage judged by GPT-4o-2024-11-20. Δ denotes average win-rate improvement over ablations, and Avg reflects the mean win rate across EN and CN.

| Comparison | EN (win/lose) | CN (win/lose) | Δ (EN / CN) | Avg Δ |
|---|---|---|---|---|
| SpecEM vs w/o Online feedback | 53.66 / 46.34 | 52.98 / 47.02 | +7.32 / +5.96 | +6.64 |
| SpecEM vs w/ Feedback (Score-based reward) | 52.17 / 47.83 | 51.63 /48.37 | +4.34 / +3.26 | +3.80 |
| SpecEM vs w/o Feedback, Top-win selection | 53.40 / 46.60 | 52.35 / 47.65 | +6.80 / +4.70 | +5.75 |

## 4.3 Analysis

**Ablation analysis.** We perform ablation studies to assess the effect of the online feedback mechanism in the verification stage. Using GPT-4 as the evaluator, we compare the full SpecEM with three variants: (a) **w/o online feedback**: Removes feedback; final output is selected solely based on verification scores. (b) **w/ score-based reward**: Replaces win count $\gamma_i$ with the normalized average verification score as the reward. (c) **w/o feedback, top-win selection**: Selects the candidate with the highest win count without reward accumulation. As shown in Figure 4, the full SpecEM outperforms all variants. Online feedback yields a 6.6-point average gain; using win count as the reward offers an additional 2.8-point improvement over score-based reward. Directly selecting the top-win segment leads to a 5.8-point drop, likely due to close or tied win counts, which are more reliable as soft rewards than as decisive selection criteria.

**Inference Latency Analysis.** We evaluate the inference efficiency of SpecEM against other ensemble methods and base models, focusing on two key metrics shown in Figure 3: (1) *First Token Latency*, the time from user input to the generation of the first token, which is crucial for interactive user experience; (2) *Total Generation Time*, the time to generate a complete response across varying output lengths. SpecEM achieves the lowest total response time among all ensemble methods, with

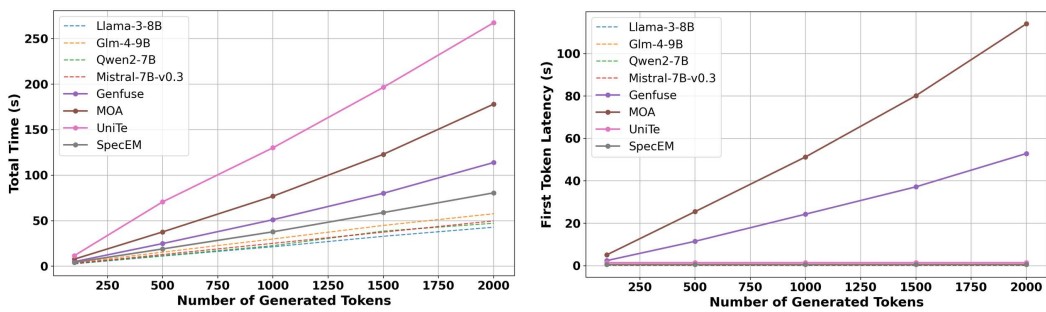

Figure 3: Comparison of inference latency performance across methods. Left: total generation time (seconds); Right: first-token latency (seconds), both plotted against the number of generated tokens.

only a 20% overhead compared to the slowest single model. This is because, under parallel inference settings, the ensemble latency is bottlenecked by the slowest model's output time.

Crucially, SpecEM maintains consistently low first token latency (under 0.6s) across all lengths, enabling fast user feedback. In contrast, methods that wait for full outputs before fusion suffer from rapidly increasing first token latency, making them unsuitable for real-time applications.

**Base Model Number Analysis.** We evaluate how SpecEM scales with the number of integrated base LLMs on the English FuseEval dataset. As shown in Figure 4, performance consistently improves as more base models are added. The improvements are more pronounced when stronger models are introduced, while even weaker models still contribute positively. These results highlight the flexibility and scalability of SpecEM, where new models can be seamlessly integrated without additional training or adaptation, making the system robust and extensible in real-world deployment.

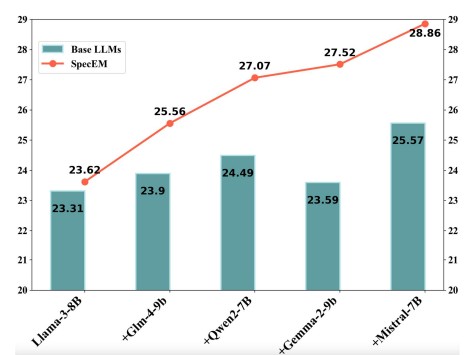

Figure 4: The variation in SpecEM's ROUGE-L score as the number of base LLMs increases. +[model] indicates the incremental addition of a specific model to the ensemble.

**Candidate Segment Length Analysis.** We study the impact of the maximum generation length $L$ of candidate segments on SpecEM's performance using the English FuseEval development set. As shown in Figure 5 (left), both BERTScore and ROUGE-L improve as $L$ increases, peaking at $L = 10$, then gradually decrease. This trend arises because shorter segments lack sufficient information, which weakens the judgment of the verification component and limits the mutual inspiration between models. Conversely, overly long segments reduce the frequency of cross-model interactions, hindering effective knowledge fusion and ultimately degrading the final output quality.

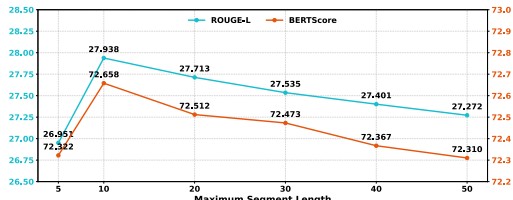
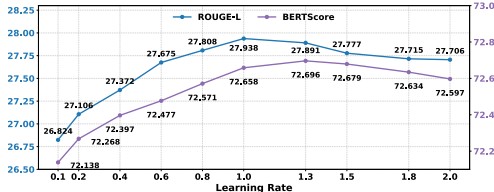

Figure 5: Performance trends of SpecEM. Left: Varying maximum candidate segment length. Right: Varying hyperparameter $\alpha$ in the online feedback.

**Online Feedback Hyperparameter Analysis.** To investigate the impact of the hyperparameter $\alpha$ in learning rate in online feedback mechanism, we vary its value from 0.1 to 2 and evaluate the

results on the English FuseEval development set. As shown in Figure 5 (right), both BERTScore and ROUGE-L scores initially increase with larger $\alpha$, stabilize around the range of 0.8–1.3, and then begin to decline. A very small $\alpha$ leads to insufficient updates, limiting the effectiveness of online feedback, while an overly large $\alpha$ causes instability and degrades performance.

## 5 Conclusion

We propose SpecEM, a training-free and plug-and-play ensemble framework for generative LLMs. SpecEM integrates model outputs through an iterative drafting-verification process at the segment level, enabling semantic collaboration across models without additional training. Further, we introduce an online feedback mechanism that dynamically adjusts each model's influence during generation based on real-time performance. Experiments across six datasets and five LLM families ranging from 7B to 72B parameters show that SpecEM can effectively coordinate multiple LLMs, demonstrating strong generalization across model scales, task types, and languages.

## 6 Limitations

Although SpecEM enables flexible integration of new models without additional training, it still faces challenges in the ensemble process. Specifically, introducing a model that performs poorly on the current task can degrade the overall performance, sometimes even falling below that of an ensemble excluding the weaker model. For instance, as shown in Table 2, on the IFEval dataset, SpecEM integrating Qwen2, GLM4, and Gemma2 underperforms compared to the ensemble of only GLM4 and Gemma2. While we adopt an online feedback mechanism to dynamically adjust each model's contribution, low-quality outputs from weaker models can still negatively affect the initial generation phase. In future work, we plan to explore rejection sampling and resampling strategies to more effectively identify and amplify stronger models in the generation process, while suppressing the influence of weaker ones.

## 7 Acknowledgments

This work was supported by the National Key Research and Development Program of China (No. 2022ZD0115301), Major Key Project of PCL via grant No. PCL2025AS11, National Natural Science Foundation of China (NSFC) via grant 62206140,62406223. Thanks for the support provided by OpenI Community (https://openi.pcl.ac.cn).

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

# A  Empirical Validation of the Correlation Between Generation and Verification Capabilities of LLMs.

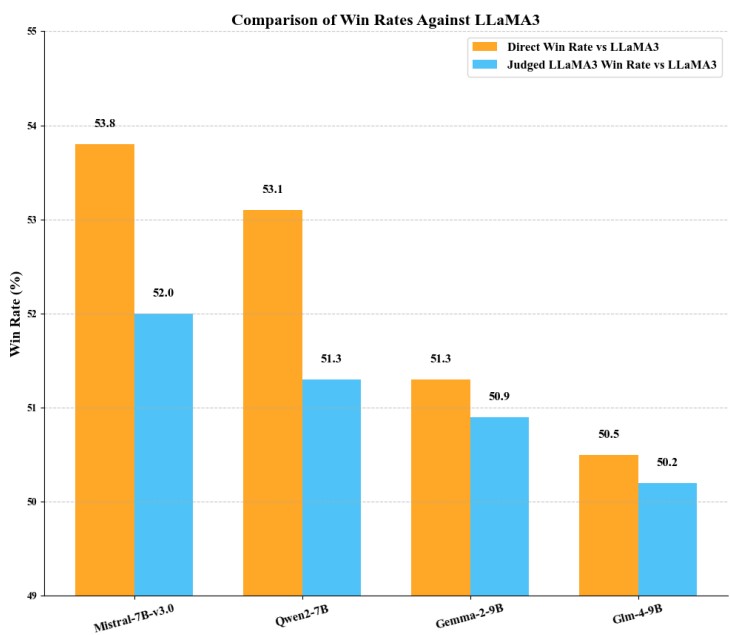

Figure 6: Win rates of four LLMs compared to Llama-3-8B-instruct, evaluated by GPT-4. Orange bars indicate generation performance (as response generators); Blue bars indicate verification performance (as candidate scorers for LLaMA). The consistent ordering supports the hypothesis that stronger generators are also stronger verifiers.

In Section 3.3, we hypothesize that models with stronger generation capabilities also possess stronger verification abilities, to conduct online feedback mechanism in SpecEM. To verify the assumption, we design an empirical experiment. Specifically, we use GPT-4 as an external evaluator, and evaluate four models, Mistral-7B-v0.3-instruct, Qwen2-7B-instruct, Gemma-2-9b-instruct, and Glm-4-9b-instruct, by comparing their responses against those generated by Llama-3-8B-instruct on FuseEval. The orange bars in Figure 6 report the win rates of these models over LLaMA, as judged by GPT-4. The results indicate a consistent performance ranking: Mistral > Qwen > Genma > GLM, with all models outperforming LLaMA.

To assess verification ability, we treat each of the four models as a verifier. For every input example, we sample four candidate responses from LLaMA. Each verifier independently scores all candidates using the method described in Section 3.2, and selects the one with the highest score. The selected responses are then evaluated by GPT-4 against the original single-response baseline from LLaMA. The blue bars in Figure 6 show the win rates of the verifier-selected responses, again consistently ranked as Mistral > Qwen > Genma > GLM.

This alignment in performance ordering across generation and verification supports our central hypothesis: models with stronger generation capabilities are more competent as verifiers. For example, Mistral not only achieves the highest generation win rate, but also, when acting as a verifier, selects the strongest responses for LLaMA, demonstrating its superior verification ability.

# B  Experimental Setup

## B.1  Dataset Details

We evaluate all the models on six datasets that represent different core capabilities of LLMs, including open-domain instruction-following, commonsense, and reasoning.

- FuseEval: We evaluate the model's instruction-response capability by constructing this category on both English and Chinese scenarios. For English, we choose the Dolly-15k [Conover et al., 2023] and Alpaca-gpt4 [Peng et al., 2023] datasets for evaluation, both of which have inputs that consist of human instructions. We select these two datasets because their response sources differ: the Dolly-15k dataset features human-provided responses, while the Alpaca-GPT4 contains responses generated by the state-of-the-art GPT-4 [OpenAI et al., 2024] model, which provides neutral reference answers to each question and can refuse to answer inappropriate or harmful ones. Using both types of responses for scoring allows us to more thoroughly compare the advantages of our ensemble system. Additionally, due to the large size of these datasets, we randomly sample portions from each to create a new test set and a development set. For Chinese, we utilize the Human-Value and Ruozb datasets from the COIG-CQIA [Bai et al., 2024] benchmark for testing. The instructions in these two datasets consist of human-posed questions, with answers provided either by humans or generated by GPT-4. The COIG-CQIA authors manually review and filter the responses, retaining only the correct answers generated by GPT-4.

- AlpacaEval 2.0 [Dubois et al., 2024] : This is an automated benchmark for evaluating large language models' instruction-following capabilities. It employs GPT-4 Preview (11/06) as an evaluator to compare model responses against a baseline (also GPT-4 Preview (11/06)), computing win rates with a length-controlled scoring mechanism to reduce verbosity bias. Given the strong performance of 24-72B foundation models, we further conducted direct comparisons between ensemble model outputs and GPT-4 generated responses.

- MMLU (5-shot) [Hendrycks et al., 2021]: A widely-used massive multitask language understanding benchmark for evaluating knowledge and commonsense reasoning across 57 subjects, including STEM, humanities, and social sciences. It assesses models' breadth of understanding across a diverse set of multiple-choice questions. x-shot refers to providing x examples as in-context during inference.

- ARC-C (5-shot) [Clark et al., 2018]: A subset of the AI2 Reasoning Challenge benchmark consisting of grade 3–9 science exam questions that require non-trivial logical reasoning. The task is formulated as a multiple-choice question answering problem.

- GSM8K (3-shot) [Cobbe et al., 2021]: A high-quality dataset of linguistically diverse grade school math word problems, curated to evaluate arithmetic reasoning capabilities. Each question requires multi-step reasoning to arrive at the correct solution.

- IFEval [Zhou et al., 2023]: A targeted benchmark for assessing instruction-following proficiency. It contains prompts with explicit directives and uses GPT-4 to evaluate how well model outputs comply with the given instructions.

## B.2 Evaluation Methods

We use a variety of metrics for different tasks, following the test scripts from the Openllm leaderboard. For FuseEval, we apply BARTScore (Bart-S) [Yuan et al., 2021], BERTScore (Bert-S) [Zhang et al., 2019], GPT4-Rank (GPT4-R) [OpenAI et al., 2024], BLEU [Papineni et al., 2002], and ROUGE (R-n) [Lin, 2004]. For multiple-choice tasks such as MMLU and ARC-C, we select the option with the highest likelihood to calculate accuracy (Acc). For the reasoning dataset GSM8K, we evaluate exact match (EM) accuracy. For IFEVAL, we rely on the evaluation files provided by the dataset creators [Zhou et al., 2023], testing under prompt-strict, instruction-strict, prompt-loose, and instruction-loose conditions. For AlpacaEval 2.0, we use the official GPT-4-based pairwise comparison framework [Conover et al., 2023], where each model's output is evaluated against a GPT-4 reference response, and the win rate is computed as the final metric. A detailed description of some evaluation metrics for FuseEval is as follows:

- BLEU (B-$n$) [Papineni et al., 2002] and ROUGE (R-$n$) [Lin, 2004] compare a generated response with a reference by calculating $n$-gram overlap. For the Chinese results, we use Jieba[2] to split the text into words before calculating these two scores.

- BERTScore [Zhang et al., 2019] (comprising Precision, Recall, and F1-score) measures the similarity between two texts based on the contextualized embedding from BERT [Devlin et al., 2019]. In this paper, we report the F1 score of BERTScore.

---

[2]https://pypi.org/project/jieba/

- BARTScore [Yuan et al., 2021] is a unified evaluator which evaluates with the average likelihood of the pretrained encoder-decoder model, BART [Lewis et al., 2019]. It can predict different scores depending on the formats of the source and target.

- The GPT4-Rank [OpenAI et al., 2024] evaluation utilizes the GPT-4o-2024-11-20 model to compare two different responses against a ground-truth response. The model will select the better of the two responses. For each test sample, we pair the responses generated by different models and have GPT-4 determine which one is superior. Since the MBR and PairRank methods do not generate new responses, we do not re-rank the responses they select from the base LLMs. Instead, we use the average rankings of the responses they select from the base LLMs to represent their GPT4-Rank. Once all comparisons are complete, we count the number of wins for each model. Based on these win counts, we rank the responses from the different models. The average ranking of each model across all data in the dataset is the value reported in our table. The evaluation instructions for GPT-4 are shown in Table 5.

- The win rate comparisons between models in this study were conducted using GPT-4o-2024-11-20 as the evaluator. Both Table 3 and Table 4 employ GPT-4o-2024-11-20 to compare outputs from different models, with the evaluation instructions for GPT-4 shown in Table 5. In Table 3: The "English FuseEval winrate" and "Chinese FuseEval winrate" metrics compare the outputs of various base models (and ensemble methods) against those generated by Qwen2_72b_instruct.

## B.3    Baselines

Since our approach has not undergone any additional training, we selecte several types of untrained baseline models for comparison with our method:

- PairRank: An English reward model introduced in the LLM-Blender [Jiang et al., 2023b], which compares candidate results generated by different LLMs and selects the best candidate as the ensemble output.

- Majority Voting [Davani et al., 2022]: each model provides a choice, and the final result is determined by the option with the most votes.

- Minimum Bayes Risk (MBR) [Freitag et al., 2023]: Selects the answer with the highest lexical similarity to other candidate answers. In this paper, we use the SimCSE [Gao et al., 2022] model to calculate the similarity between candidate responses.

- Generation Fusion (GF) [Jiang et al., 2023b]: Uses the outputs of other models as context, passing them to a new model, which generates a response based on this context. In our implementation, Mistral-7B-v0.3 is employed as the final model for the 7B–9B scale integration, and Mistral-24B-instruct-2501 for the 24B–72B scale integration, based on their performance advantages.

- Mixture-of-agents (MOA) [Wang et al., 2024]: Multi-layer fusion is applied, where the outputs of all base models are concatenated and fed back into the models, with an aggregator outputting the final result. In this work, we adopt the stronger-performing Mistral-7B-v0.3 as the aggressor in the 7B–9B scale integration, and the better-performing Mistral-24B-instruct-2501 as the aggressor in the 24B–72B scale integration. The fusion process is repeated three times, consistent with the original MOA methodology.

- Unite [Yao et al., 2025]: Constructs a new union vocabulary by combining the vocabularies of multiple models to include all tokens from each model.

## C    Case Study

Table 7 presents a case from the SpecEM workflow where the user's request is "Write a simile to describe a person who is hard-working." The reasoning process goes through four iterations, and the verify model's selection of the best candidate is not always from the same model. In the first round, the best candidate is generated by Qwen2. In the second round, Mistral, after receiving Qwen2's output from the previous round, is inspired and generates a response that better meets the user's needs, as using "farmer" to describe a hard-working person is inappropriate. Additionally, the table shows

Table 5: The template used for GPT-4 compares two models' responses.

| | |
|---|---|
| **Template** | Instruction:
${instruction}

Ground-Truth Response:
${Truth response}

Model A:
${A response}

Model B:
${B response}

Given the User's Instruction and Ground-Truth response above, please compare the two Model's responses.
You only have 2 choices to output:
If you think A is better, please output: 1. A is better
If you think B is better, please output: 2. B is better
Output your choice below: |
| **Comparison Option** | 1. A is better
2. B is better |

that through mutual inspiration between models, the final response more closely matches the user's expectations, thereby improving the overall quality of the reply.

# D   Discussion on Segment Length

SpecEM employs fixed-length candidate segments for model evaluation and routing, which simplifies the computation process but may occasionally cause semantic truncation, particularly when segment boundaries intersect with meaningful linguistic or contextual units. This truncation can weaken semantic coherence and reduce the accuracy of optimal segment selection. In contrast, adaptive segmentation approaches such as NUDGING Fei et al. [2025] dynamically determine token boundaries based on uncertainty estimation across model families. However, these methods assume a shared vocabulary and tokenization scheme, which restricts their applicability in heterogeneous large-model scenarios where model architectures and vocabularies differ. Future work could explore adaptive or semantics-aware segmentation mechanisms that maintain context completeness while remaining compatible with multi-model routing frameworks.

| | | User instruction: Write a simile to describe a person who is hard-working. | | |
|---|---|---|---|---|
| Input for each round | Qwen2 Generation | Mistral Generation | Glm-4 Generation | Best Candidate |
| **Round 1:**
Write a simile to describe a person who is hard-working. | A person who is hard-working is like a determined | This person is as dedicated to their work as | A hard-working person is like a well | A person who is hard-working is like a determined |
| **Round 2:**
Write a simile to describe a person who is hard-working. A person who is hard-working is like a determined | farmer, tirelessly cultivating the land and nurturing their crops | river, steadily carving its path through the | horse pulling a heavy cart through a long journey. | river, steadily carving its path through the |
| **Round 3:**
Write a simile to describe a person who is hard-working. A person who is hard-working is like a determined river, steadily carving its path through the | rockiest of landscapes. <\|im_end\|> | mountains. | rocks and obstacles in its way, never giving up | rocks and obstacles in its way, never giving up |
| **Round 4:**
Write a simile to describe a person who is hard-working. A person who is hard-working is like a determined river, steadily carving its path through the rocks and obstacles in its way, never giving up | until it reaches the sea. | until it reaches its destination. | . <\|user\|> | until it reaches its destination. |

Overall Output: A person who is hard-working is like a determined river, steadily carving its path through the rocks and obstacles in its way, never giving up until it reaches its destination.

Figure 7: Case study of SpecEM integrating the base LLMs Qwen2, Mistral, and GLM-4. The Best Candidate is the top-ranked option determined by the verify component and is directly presented to the user. $< |im\_end| >$, $< /s >$, and $< |user| >$ are special end tokens for the three base LLMs, and generation halts when the best candidate includes any of these end tokens.

## NeurIPS Paper Checklist

The checklist is designed to encourage best practices for responsible machine learning research, addressing issues of reproducibility, transparency, research ethics, and societal impact. Do not remove the checklist: **The papers not including the checklist will be desk rejected.** The checklist should follow the references and follow the (optional) supplemental material. The checklist does NOT count towards the page limit.

Please read the checklist guidelines carefully for information on how to answer these questions. For each question in the checklist:

- You should answer [Yes] , [No] , or [NA] .
- [NA] means either that the question is Not Applicable for that particular paper or the relevant information is Not Available.
- Please provide a short (1–2 sentence) justification right after your answer (even for NA).

**The checklist answers are an integral part of your paper submission.** They are visible to the reviewers, area chairs, senior area chairs, and ethics reviewers. You will be asked to also include it (after eventual revisions) with the final version of your paper, and its final version will be published with the paper.

The reviewers of your paper will be asked to use the checklist as one of the factors in their evaluation. While "[Yes] " is generally preferable to "[No] ", it is perfectly acceptable to answer "[No] " provided a proper justification is given (e.g., "error bars are not reported because it would be too computationally expensive" or "we were unable to find the license for the dataset we used"). In general, answering

"[No] " or "[NA] " is not grounds for rejection. While the questions are phrased in a binary way, we acknowledge that the true answer is often more nuanced, so please just use your best judgment and write a justification to elaborate. All supporting evidence can appear either in the main paper or the supplemental material, provided in appendix. If you answer [Yes] to a question, in the justification please point to the section(s) where related material for the question can be found.

IMPORTANT, please:

- **Delete this instruction block, but keep the section heading "NeurIPS Paper Checklist",**
- **Keep the checklist subsection headings, questions/answers and guidelines below.**
- **Do not modify the questions and only use the provided macros for your answers**.

