# OpenReview forum: "SpecEM: Training-Free LLM Ensembling via Iterative Drafting, Verification, and Online Feedback"
_NeurIPS.cc/2025/Conference — NeurIPS 2025 poster_

### Official Review · Reviewer_KAZA · 2025-06-19

**Clarity:** 3
**Significance:** 2
**Originality:** 2
**Rating:** 4
**Confidence:** 4

**Summary:**

The key idea of this paper is to use speculative decoding (which was used earlier for acceleration) to ensemble multiple LLMs at inference time. It targets an ensemble-while-generation approach for better inference latency as opposed to generate-then-ensemble.  The approach first drafts (all LLMs generate candidates as “segments” with predefined maximum length) and then verifies (all LLMs mutually evaluate all generated candidates in parallel; produce MxM scores for M ensemble models), followed by an online learning mechanism to adjust the weight for better-performing models over time dynamically. Next, the top-ranked segment is selected as the best output and forwarded to all models for subsequent rounds of drafting-verification.
The model’s evaluation score is the average of the logits produced by the model over the tokens in the segment. The final score for each candidate is a weighted sum of the scores from all verifier models, where scaler weights are dynamically updated by the proposed online feedback mechanism based on a simple reward (counting the number of times a model candidate is scored higher than another candidate by a third model).

**Questions:**

Q1) Focusing on widely known benchmarks of MMLU and GSM8k in Table 2, the proposed method by ensembling two or three LLMs (including Gemma-2) shows marginal score improvement (max 1.5%) compared to the best LLM, which is Gemma-2. This raises the concern whether simpler methods, such as self-consistency with majority voting on Gemma-2, would reach similar scores without necessarily using different LLMs?

Q2) Please elaborate on this statement:
> Despite using only several 7B–9B LLMs, SpecFuse performs comparably to 70B-scale single models while remaining more parameter efficient.

It seems that it doesn’t quite hold, because 70B-scale LLMs are consistently outperformed in the first place by smaller base LLMs in Table 1 and Table 3 (except on BARTScore on English Scenario in Table 1), and there are no large LLMs for Table 2.

Q3) It would be great to see the impact of the method on more recent benchmarks such as GSM-hard, or MATH-500

**Ethical Concerns:**

["NO or VERY MINOR ethics concerns only"]

**Final Justification:**

Thoughtfully addressed the fusion concept and showed willingness to revise the title to more clearly reflect the underlying technique, reducing potential confusion. I also appreciate the additional self-consistency results, which effectively demonstrate the cost-benefit trade-off of ensembling. These clarifications and additions have strengthened the work, and as a result, I am happy to increase my score to 4.

**Limitations:**

No limitation is discussed. One possibility is to open up the limitation of he method compared to larger LLMs, or smaller LLMs when using a simple test-time scaling approach, on the studied benchmarks.

**Quality:**

2

**Strengths And Weaknesses:**

Strengths: The paper is clearly written, and the proposed method is positioned well wrt prior art in ensemble-while-generation. The online feedback mechanism seems to be novel in this context.

The weakest conceptual point is that this method is not a fusion in any form (as opposed to what has been called in the paper) but simply a selection (the highest-scoring candidate is selected as the best output). Hence, the title, abstract, and intro are misleading and should be adjusted accordingly.

For technical weakness, see questions.

---

> ### Author Rebuttal · Authors · 2025-07-31
>
> Dear Reviewer KAZA,
>
> **Thank you for your thoughtful and thorough review of our manuscript. We truly appreciate the time and effort you dedicated to understanding our work, and your constructive comment. Below, we respond to each of the suggestions you raised. We hope this addresses your concerns and helps improve your scoring of our work.**
>
> **[weakness1:Concerns regarding fusion concepts]**
>
> * Thank you very much for your insightful comments. We fully understand your concern, as we mentioned selecting the best model candidate at each iteration. We assume that by “fusion,” you refer to combining all models’ votes or weighted scores to select the optimal token at each round—that is, token-level fusion. However, we would like to clarify that in our work, fusion refers to fusion between segments. Although within each segment we select the single best candidate from one model, the final output is composed of segments generated by different models, **representing segment-level fusion**, which ensures longer semantic coherence and fewer fusion points. Furthermore, during each iteration’s segment generation, the best segment is broadcast to all models, meaning that the historically best segments chosen by other models are fused as semantic context in the current round. These models are thus inspired by the historical best segments to continue generation.
>
> We sincerely hope the above explanation clarifies your concerns regarding the concept of fusion. If you still have concerns, we would be happy to further discuss the appropriateness of using the term “fuse” and are open to revising the wording to make the method description more precise and rigorous.
>
>
> **[Question1: whether simpler methods, such as self-consistency with majority voting on Gemma-2, would reach similar scores without necessarily using different LLMs?]**
>
> * We sincerely appreciate your constructive suggestions. Following your suggestion, we conducted additional self-consistency experiments with Gemma-2 under the same experimental setup as in Table 2, specifically on the MMLU and GSM8K datasets. For each sample, Gemma-2 was prompted to generate three predictions, and the final answer was selected by majority voting. If all three predictions differed, one was randomly chosen. The experimental results are as follows:
>
>   | | MMLU  | GSM8k |
>   |------------------------------|-------|-------|
>   | Gemma-2                      | 71.51 | 77.26 |
>   | Gemma-2（ self-consistency=3） | 71.74 | 77.75 |
>
>
>   As shown in the table, Gemma-2 (self-consistency) achieves an improvement of +0.23 on MMLU and +0.39 on GSM8K compared to the original Gemma-2. While this demonstrates some gains under the same computational budget, SpecFuse achieves a much larger improvement of +1.5.
>
>   Moreover, the applicability of the self-consistency method is relatively limited. In open-domain text generation tasks, the system typically cannot determine in advance which domain a user query belongs to, nor which model performs best for that particular query. As a result, it is difficult to apply self-consistency, an approach that relies on multiple independent generations and aggregation in such scenarios, especially in non-multiple-choice tasks where standard answers are unavailable. This highlights the strength of ensemble methods: by integrating multiple heterogeneous models, SpecFuse leverages their complementary strengths to enhance the stability of system outputs in open-ended tasks.
>
>
> **[Question2: Elaborate on the statement: "Despite using only several 7B–9B LLMs, SpecFuse performs comparably to 70B-scale single models while being more parameter efficient."]**
>
> * Thank you for your question. We would like to clarify a possible misunderstanding that may have arisen from the pink and blue markings in the table. As described in the caption of Table 1, the blue marks indicate the best-performing 7–9B base LLMs (excluding 70B-scale larger LLMs), while the pink marks indicate the best results overall, including base models, larger models, and ensemble methods. Generally, aside from Mistral-7B-Instruct-v0.3, which achieves performance comparable to LLaMA-3-70B-Instruct, larger LLMs consistently outperform base LLMs. We have extracted the best-performing models from both the base model group and the larger model group, as shown below:
>
>   | Model | ROUGE-1↑ | ROUGE-2↑ | ROUGE-L↑ | BLEU↑   | BARTScore↑ | BERTScore↑ | GPT4-Rank↓ |
>   |--------------------------|----------|----------|----------|---------|------------|------------|------------|
>   |  | | | | English ||||
>   | Mistral-7B-instruct-v0.3 | 27.75    | 10.75    | 25.57    | 4.82    | -2.94 | 71.88| 7.62  |
>   | Mixtral-8x7B-instruct    | 29.04    | 12.25    | 26.75    | 4.08    | -2.81 | 72.19 | 4.16|
>   ||  |    |  | Chinese | |  | |
>   | Glm-4-9B-instruct | 30.88    | 8.71     | 20.42    | 4.47    | -4.33      | 70.25      | 5.18       |
>   | Qwen2-72B-instruct | 31.44    | 8.97     | 22.48    | 4.88    | -4.32      | 70.65      | 3.63       |
>
>   The upward arrow ↑ means higher is better, and the downward arrow ↓ means lower is better.
>
>    As shown in the table, except for the BLEU score on the English dataset, the larger models outperform the base models on all other metrics, which supports the validity of our statement.
>
> * Moreover, the comparison with larger models is intended as a reference rather than our primary focus. It aims to illustrate that, for some generation tasks, combining multiple smaller models can yield competitive performance compared to much larger models with more parameters and training data. Our main goal remains to compare against previous ensemble baselines.
>
> Thank you for your question, we hope the above response addresses your concern.
>
>
> **[Question3: It would be great to see the impact of the method on more recent benchmarks such as GSM-hard, or MATH-500
> ]**
>
> * Thank you very much for your constructive suggestion. Based on your advice, we added experiments on these two datasets, where SpecFuse integrates three base LLMs: GLM-4, Gemma-2, and Qwen2. All experiments were conducted under the same zero-shot setting. The results are as follows:
>
>   | Model    | GSM-hard | MATH-500 |
>   |----------|----------|----------|
>   | GLM-4| 50.03| 48.00 |
>   | Gemma-2| 55.02| 49.80|
>   | Qwen-2 | 52.55| 51.40|
>   | SpecFuse | 56.10| 53.40 |
>
>
>   The results show that SpecFuse achieved the highest accuracy, outperforming the best single model, Gemma-2, by 1.38 points on GSM-Hard, and surpassing the best single model, Qwen2, by 2 points on Math-500.
>
> Thank you again for your valuable suggestion. We hope our response helps resolve your confusion.
>
>
> **[Limitations:No limitation is discussed.One possibility is to open up the limitation of the method compared to larger LLMs, or smaller LLMs when using a simple test-time scaling approach.]**
>
> * Thank you for your thoughtful suggestion. As noted in Appendix E of the original paper, we discuss some limitations of our method, particularly the potential impact when there is a significant performance gap among base models.
> Regarding your point about comparisons with larger LLMs or smaller ones using simple test-time scaling, we will include a detailed discussion on this aspect in the revised version of the paper.
>
> Thank you for your suggestion, we hope the above response addresses your concern.
>
>
> **Thank you for taking the time to read our comments. If we have addressed your concerns, we sincerely hope you can reconsider and increase your score. If you have any other questions, we look forward to further discussions with you. We sincerely await your reply.**

---

> ### Comment · Reviewer_KAZA · 2025-08-01
> **fusion concepts and cost of self-consistency**
>
> Thanks for providing the rebuttal, explanations, and extra results. However, two main issues remain as discussd in the following.
>
> > Although within each segment we select the single best candidate from one model, the final output is composed of segments generated by different models, representing segment-level fusion
>
> Independent of the level of ensemble-while-generating, I still cannot find a fusion function. At the segment-level, there is no fusion function but simply a selection that takes the highest-scoring candidate segment as the best output. Note that selection itself is not a fusion function. Selection is a process of choosing specific elements from a set (here, a set of candidate segments), while fusion is a process of combining or merging different elements. Precisely, the proposed method is a **segment-level selection among an ensemble.**
>
> >While this [using self-consistency = 3] demonstrates some gains under the same computational budget, SpecFuse achieves a much larger improvement of +1.5.
>
> Thanks for providing the self-consistency results. Why only 3? I would suggest providing a wide range of self-consistency in the order of 10-50 as in the literature. Note that they don’t have “the same computational budget” because the self-consistency avoids the prefilling cost which could be major by promoting and keep conditioning different models.

---

> ### Author Response · Authors · 2025-08-03
> **Response to Reviewer KAZA (Round 2)**
>
> Dear Reviewer KAZA,
>
> **We sincerely appreciate you taking the time to review our comments and for your thoughtful reconsideration of our work. Below, we provide further clarification on the two issues you raised. We hope this addresses your concerns and helps improve your evaluation of our work.**
>
>
> **[Question 1：Fusion concepts]**
>
> * Thank you very much for your reply and for taking the time to continue discussing with us.  We recognize that a potential reason for the differing understanding of “fusion” may stem from our inspiration by the GENFUSER (Generative Fusion) method proposed by Jiang et al [1], which is one of the earliest studies in the area of LLM ensembling.
>
>   This type of method uses the outputs of other models as context and feeds them to a new model, allowing the new model to generate new outputs by referencing this knowledge. We were deeply influenced by this approach and described the way multiple models generate new segments by referring to the best outputs from previous rounds as fusion. However, the concept of “fusion” we use here differs somewhat from the fusion concept you mentioned.
>
>   We sincerely apologize for any confusion caused by our differing understanding of this concept. To make the presentation in the paper more precise, we are willing to revise it. We would like to change the title to **SpecEM**: Training-Free LLM Ensembling via Iterative Drafting, Verification, and Online Feedback, where EM stands for Ensemble, essentially indicating that our method is an ensemble approach. We will also correct the parts of the paper that refer to “fuse.”
>
> We sincerely thank you once again for taking the time to discuss the concept of “fuse” with us in detail. Your guidance makes our paper more rigorous.
>
>
> **[Question 2: Conduct more self-consistency experiment]**
>
> * Regarding the use of self-consistency = 3 in the experiment mentioned in our response to Q1 of the rebuttal, this setting was chosen because Table 2 involves three different models, which corresponds to three independent inferences on the same question. Therefore, we used the same number of inference times for comparison.
>
> * We sincerely appreciate your valuable suggestion to explore a wider range of self-consistency (10–50). Following your recommendation, we have promptly conducted the additional experiments, and the results are presented in the table below:
>
>   |       | MMLU  | GSM8k |
>   |-------------------------------|-------|-------|
>   | Gemma-2                       | 71.51 | 77.26 |
>   | Gemma-2（ self-consistency=3）  | 71.74 | 77.75 |
>   | Gemma-2（ self-consistency=10） | 71.83 | 77.87 |
>   | Gemma-2（ self-consistency=20） | 71.83 | 77.91 |
>   | Gemma-2（ self-consistency=30） | 71.81 | 77.90 |
>   | Gemma-2（ self-consistency=40） | 71.82 | 77.93 |
>   | Gemma-2（ self-consistency=50） | 71.83 | 77.93 |
>
>   The experimental results show that when the number of inference attempts in the self-consistency experiment reaches 20, the performance basically stabilizes, yielding improvements of 0.32 and 0.65 respectively, which is lower than the +1.5 increase achieved by SpecFuse. This phenomenon occurs because the self-consistency method mainly alleviates random biases in the generation process through multiple sampling but cannot compensate for the model's inherent weaknesses (such as those in Gemma-2), which is precisely where model integration has its advantage.
>
> Thank you again for your valuable suggestions. The addition of these experimental results and discussions will also help strengthen our work.
>
>
> **Thank you for taking the time to read our comments. If we have addressed your concerns, we sincerely hope you can reconsider and increase your score to a positive one. If you have any other questions, we look forward to further discussions with you. We sincerely await your reply.**
>
>
> References：
>
>
> [1] LLM-BLENDER: Ensembling Large Language Models with Pairwise Ranking and Generative Fusion, ACL 2023

---

> > ### Comment · Reviewer_KAZA · 2025-08-03
> >
> > Thank you for thoughtfully addressing the fusion concept and for your willingness to revise the title to more clearly reflect the underlying technique, reducing potential confusion. I also appreciate the additional self-consistency results, which effectively demonstrate the cost-benefit trade-off of ensembling. These clarifications and additions have strengthened the work, and as a result, I am happy to increase my score to 4.

---

> > > ### Author Response · Authors · 2025-08-03
> > >
> > > Dear Reviewer KAZA,
> > >
> > > We are truly grateful that our clarifications were able to address your concerns. Your updated score and encouraging feedback mean a great deal to us, and we sincerely appreciate your support and recognition of our work.
> > >
> > > Warm regards,
> > >
> > > All Authors of the Paper

---

### Official Review · Reviewer_VDu3 · 2025-07-01

**Clarity:** 3
**Significance:** 3
**Originality:** 3
**Rating:** 5
**Confidence:** 4

**Summary:**

This paper introduces SpecFuse, a novel, training-free framework for ensembling large language models (LLMs) at inference time. Unlike previous ensemble methods, SpecFuse enables segment-level collaboration among multiple LLMs by iteratively performing two main stages: drafting (where each model generates a candidate segment) and verification (where all models evaluate each other's segments). A central innovation is the online feedback mechanism, which dynamically updates each model’s influence via multiplicative weight updates, rewarding models whose candidates are more often preferred by peers. The authors show strong empirical results across six benchmarks and five model families (from 7B to 72B parameters), demonstrating that SpecFuse consistently outperforms both single LLMs and state-of-the-art ensemble baselines in instruction-following, commonsense, and reasoning tasks. The method is entirely plug-and-play, requiring no additional training or fusion modules.

**Questions:**

1. **On Weighting Dynamics and Diversity:**
   Given the online feedback mechanism’s tendency to upweight high-performing models, is there a risk of converging to a single-model regime, thus losing the core ensembling advantage? Have you considered or tested approaches that explicitly preserve some minimum influence for weaker or specialist models, such as entropy regularization or soft lower bounds on weights? Under what conditions do you observe diversity collapse in practice?

2. **Failure Modes of Generation-Verification Alignment:**
   Can you elaborate on cases where a model may be strong at generation but unreliable at verification, or vice versa? For example, might there be scenarios where a model is verbose and persuasive but inaccurate, or vice versa? How robust is SpecFuse to such misalignments, and could alternative reward/weighting schemes improve resilience?

3. **Calibration and Scoring Metric Choice:**
   Your method uses average logit (normalized across models) as the primary segment scoring metric in the verification stage. This appears to implicitly assume that all models are reasonably calibrated, such that a higher logit truly reflects better output quality. If a model is systematically miscalibrated (overconfident or underconfident), could this bias the verification and ensemble process? Have you considered or compared alternative scoring functions (e.g., mean log-probability, entropy, or learned reward models), and do you have empirical evidence that average logit is the most robust choice? A deeper discussion or ablation here would be valuable.

4. **Long-Range Dependencies and Semantic Consistency:**
   Does segment-level iterative generation ever result in local coherence but global inconsistency or contradictions? Have you observed failure cases where context is not preserved well across rounds, and if so, are there modifications that can mitigate this?

**Ethical Concerns:**

["NO or VERY MINOR ethics concerns only"]

**Final Justification:**

Thanks authors for the thoughtful response and additional experiments. All my concerns are address, and I am thus raising my score and vote for acceptance.

**Limitations:**

Yes, but I think the paper would still benefit from an expanded discussion of both technical and societal limitations. In particular, a more detailed analysis of when and why the weighting mechanism may reduce diversity, deeper consideration of calibration and scoring metric choice, and a clearer articulation of possible negative societal impacts and mitigation strategies would strengthen the work. I think it's worth adding some discussions on the risks of value amplification or emergent behaviors when ensembling diverse LLMs. For example, could combining LLMs with divergent or poorly-aligned objectives lead to unpredictable or harmful outputs? What safeguards would you recommend? Not saying this should be a focus of the paper or be a basis for accept/reject decisions, but such discussions could be helpful.

**Paper Formatting Concerns:**

Yes

**Quality:**

3

**Strengths And Weaknesses:**

### Strengths

- **Originality:**
  SpecFuse innovates by combining speculative decoding concepts with model ensembling, creating an entirely training-free and adaptable approach for integrating multiple LLMs. The segment-level iterative collaboration is a significant departure from generate-then-ensemble and ensemble-while-generation paradigms.

- **Quality:**
  The methodology is clearly described, with thorough ablations and analysis. The attention to technical details, such as efficient mutual verification (with verify-in-line attention masks and position id remapping), is impressive. The authors also provide a thoughtful investigation into hyperparameters and scalability.

- **Significance:**
  Ensembling is a key path to robust and safe LLM deployment, and a training-free, plug-and-play solution with strong empirical gains is both timely and impactful. The consistent improvements across diverse benchmarks and model scales underscore broad utility.

- **Clarity:**
  The paper is clearly written, well-organized, and includes helpful figures to clarify the workflow (e.g., the iterative drafting-verification-feedback loop). The empirical validation of the link between generation and verification ability is a valuable addition.

- **Efficiency:**
  Compared to previous methods, SpecFuse achieves lower first-token latency and faster inference, making it more suitable for real-time or interactive applications.

### Weaknesses

- **Potential for Dominance by a Single Model:**
  The multiplicative weighting scheme, while adaptive, risks allowing the "strongest" model to dominate over time, particularly in settings where one model is consistently preferred. This could diminish the diversity benefits of ensembling, especially in heterogeneous task environments where different models have specialized strengths. The authors mention and partially investigate this, but further analysis and possible mitigation would strengthen the work.

- **Assumption of Generation-Verification Alignment:**
  The core assumption is that models strong at generation are also reliable as verifiers. While empirically supported, there may be edge cases (e.g., adversarial or highly specialized tasks) where this alignment breaks down. More discussion on potential failures of this assumption would be valuable.

- **Implicit Assumption of Calibration in Logit-Based Scoring:**
  The segment verification process relies on the average logit (or normalized average logit) for scoring candidates. This approach implicitly assumes that all models are at least somewhat well-calibrated: that is, their token-level logits meaningfully reflect output quality and are comparable across models. However, if a model is systematically overconfident or underconfident, or if models are miscalibrated in different ways, the segment selection mechanism might be biased or unreliable—even with normalization. The paper does not discuss the sensitivity of the method to calibration errors, nor does it compare to alternative scoring functions (e.g., mean log-probability, entropy, or reference-based metrics). Deeper analysis or empirical study of this point would help clarify whether average logit is the best possible choice for robust ensembling.

- **Limited Discussion of Negative Societal Impacts:**
  The paper focuses primarily on technical contributions and empirical results. Broader discussion on societal implications, such as ensembling models with conflicting value systems or potential misuse in amplifying harmful outputs, is not addressed in depth.

---

> ### Author Rebuttal · Authors · 2025-07-31
>
> Dear Reviewer VDu3,
>
> **Thank you for your thoughtful and thorough review of our manuscript. We truly appreciate the time and effort you dedicated to understanding our work. Below are our responses to your comments, which we hope address your concerns and and helps further improve your evaluation of our work.**
>
> **[Weakness1&Question1: Potential for Dominance by a Single Model and Weighting Dynamics on Diversity:]**
>
> * Thank you for your suggestion. We would like to clarify your concern regarding the diversity of the ensemble. We set the minimum reward count to 1, ensuring it is never zero (i.e., the numerator in Equation 9 is at least 1). This design aims to allow dominant models to leverage their strengths while ensuring that weaker models still retain some influence, thereby preserving the core advantage of the ensemble.
>
> * Furthermore, we calculate the percentage of times each base model produces the best candidate among all outputs on the FuseEval English set, as shown in the table below.
>
>   | Llama-3-8B | Mistral-7B | Qwen2-7B | Glm-4-9B | Gemma-2-9B |
>   |------------|------------|----------|----------|------------|
>   | 15.09%| 27.10%| 22.21%| 18.78% | 16.83% |
>
>   Experimental analysis shows that when the performance gap between models is not significant, thus allowing effective ensembling, the diversity advantage of the ensemble is preserved. Each model contributes a certain proportion of candidates that are selected as the best.
>
> We sincerely thank you for your valuable suggestion, which greatly helps us improve the quality of the paper. We will include this additional experiment in the appendix.
>
> **[Weakness2&Question2:Assumption and Failure Modes of Generation-Verification Alignment]**
>
> * Thank you for question.
> Experiments presented in Figure 6 of Appendix A  show that models with stronger generation capabilities also tend to perform well on verification tasks. While this supports the alignment assumption, we have carefully considered the edge cases you mentioned, where generation and verification may diverge.
>
>   We hypothesize an extreme scenario in which a strong verifier model, due to do_sample, occasionally selects a low-probability token, leading to generation failure. In such cases, misalignment between generation and verification may occur. However, this misalignment is more likely to arise when a model acts solely as a generator. As a verifier, such failures are less likely due to multiple models verifying in parallel, which helps correct occasional errors from individual generators.
>
> Thank you again for your valuable suggestion. We will include this discussion as part of Appendix A.
>
> **[Weakness3&Question3:Implicit Assumption of Calibration in Logit-Based Scoring and Metric Choice]**
>
> Thank you for your question. We would like to address your concern with the following explanation.
>
> * First, the score of each candidate is computed by summing the scores assigned by all models. Even if a single model is overconfident (assigning consistently high scores) or underconfident (assigning consistently low scores), the normalization ensures that all models operate on a consistent scale. In this way, the influence of miscalibrated models is mitigated, as other models can effectively calibrate the final score by providing more balanced evaluations.
>
>   For example, suppose models a b and c generate candidate segments in one round. Model a assigns scores line_a = [2.3 2.6 3.2] for candidates a b c, model b gives line_b = [1.4 1.6 2.2], and model c gives line_c = [0.6 0.3 0.5]. Since these raw scores differ in scale, we normalize each model’s scores over the candidate set, resulting in line_a = [0.40 0.32 0.28], line_b = [0.42 0.31 0.27], and line_c = [0.27 0.34 0.40].Without online feedback, candidate a’s score = 0.40 + 0.42 + 0.27 = 1.09 > candidate c’s score = 0.28 + 0.27 + 0.40 = 0.95
> This shows that even if model c is overconfident, its influence can be corrected by the other models.
>
> * Second, logit-based scoring is a widely adopted unsupervised approach for evaluating outputs, commonly used in metrics such as perplexity and confidence estimation. Since language models inherently generate tokens based on logits, inaccurate logits would also lead to poor generation quality.
> We provide consistency experiments in Appendix A, showing that models perform consistently as both generators and verifiers, those with better generation quality tend to produce more accurate scores. For biased or low-quality models, their poor generation also reflects unreliable scoring. Our online feedback mechanism downweights such models in the verifier role, leading to more accurate overall scoring.
>
> * Additionally, in the scoring metric comparison, we evaluated several alternatives. Specifically, we applied softmax to the logits to obtain probabilities, then took the logarithm to compute the mean log-probability of the generated sequence. This was compared with directly using token-level logits as in our main approach. Experimental results on the FuseEval English dev set are as follows:
>
>   |Metric| rougel | bertscore |
>   |-------------------|--------|-----------|
>   | logits | 27.94| 72.66|
>   | log-probabilities | 27.47 | 72.32 |
>
>   The experimental results show that directly using the average of token-level logits yields better performance.
>
> We sincerely appreciate your thoughtful question and hope that our response has clarified the issue.
>
> **[weakness4&Limitation： Discussion of Negative Societal Impacts]**
>
> We sincerely thank the reviewer for raising these important concerns regarding both technical and societal limitations.
>
> * Our current submission primarily focuses on technical design and empirical evaluation, we fully recognize the relevance of broader implications, especially when ensembling large language models (LLMs) with diverse or potentially misaligned objectives. In future versions of this work, we are committed to including a dedicated section on ethical and societal considerations. This will include a more detailed analysis of when and how our weighting mechanism may inadvertently reduce output diversity and a reflection on the risks of value amplification or emergent behaviors when combining heterogeneous LLMs. We also intend to highlight the need for proactive safeguards, such as post-ensemble moderation, and to call on the community to further investigate robustness and safety in multi-model ensembling settings.
>
> We appreciate the reviewer’s thoughtful feedback, which will help us broaden the impact and responsibility of our research.
>
> **[Question: Long-Range Dependencies and Semantic Consistency]**
>
> * Thank you for your question regarding long-range dependencies and semantic consistency. In our observations, such issues rarely occur. This is because each current segment is selected from candidate segments based on a voting mechanism, using a context composed of previously selected segments. The scoring is based on logits, which are strongly correlated with model perplexity, to evaluate each candidate’s semantic relevance to the preceding text and its overall fluency.
> If a candidate segment contains semantic inconsistencies, leading to higher perplexity, it is unlikely to be selected.
>
> * To verify this, we constructed a case to simulate such a scenario. Specifically, we assembled four candidates—three normal outputs from Qwen2, Mistral, and GLM-4, and one manually crafted noisy candidate with semantic inconsistency. As shown in the table below, during each round of the ensembling process, SpecFuse consistently selected candidates from Qwen2, Mistral, or GLM-4, while assigning increasingly lower scores to the human-crafted noisy candidate. This demonstrates that SpecFuse effectively avoids semantically inconsistent candidates during the verification stage.
>
>   | | Qwen2| Mistral| Glm-4| Human   Noise| Best|
>   |-----------------|----------------------------------------------------|------------------------------------------------|---------------------------------------------------|------------------------------------------------------|----------------------------------------------------|
>   | Round   1 Text  | A   person who is hardworking is like a determined | This   person is as dedicated to their work as | A   hard-working person is like a well | Title:   The Cowardly Man | A   person who is hardworking is like a determined |
>   | Round   1 Score | 0.292 | 0.2535| 0.259| 0.1955||
>   | Round   2 Text  | seed   that refuses to give up until it grows | river,  carving its path through the | oak,   roots sunk deep in the earth | slacker   waiting for success without doing anything | river,   carving its path through the  |
>   | Round   2 Score | 0.2609 | 0.312 | 0.2646  | 0.1717 | |
>
> * Furthermore, your concern could be addressed by generating along the top-k best segments at each round, rather than using the current top-1 selection strategy adopted in this work. This is essentially a beam search approach, which explores a broader search space to approximate a more globally optimal solution. However, this also comes with increased computational cost.
> Thank you for your suggestion，we plan to include beam search as an optional decoding strategy in future versions of our code, allowing users to choose the approach that best suits their computational resources.
>
> **Thank you for taking the time to read our comments. If we have addressed your concerns, we sincerely hope you can reconsider and improve your evaluation of our work. If you have any other questions, we look forward to further discussions with you. We sincerely await your reply.**

---

> > ### Comment · Reviewer_VDu3 · 2025-08-06
> >
> > Thanks authors for the thoughtful response and additional experiments. All my concerns are address, and I am thus raising my score and vote for acceptance.

---

> > > ### Author Response · Authors · 2025-08-08
> > >
> > > Dear Reviewer VDu3,
> > >
> > > Thank you for taking the time to review our response.  We are truly grateful for your decision to raise the score and sincerely thank you for your support and recognition of our work.
> > >
> > >
> > > Best wishes,
> > >
> > > All Authors of the Paper

---

### Official Review · Reviewer_tSy8 · 2025-07-01

**Clarity:** 3
**Significance:** 3
**Originality:** 3
**Rating:** 4
**Confidence:** 4

**Summary:**

This paper proposes a training-free, plug-and-play LLM ensemble method, SpecFuse, that integrates multiple models' outputs by iteratively coordinating drafting and verification with an online weights feedback mechanism to adjust the influence of each model based on their contributions. By interleaving drafting and verification at the segment level, it keeps first-token latency near that of a single model while adding only ~20 % runtime overhead. Tested on six instruction-following and reasoning benchmarks with 7-72 B models, SpecFuse outperforms base models and state-of-the-art ensemble baselines.

**Questions:**

1. Can you explain why to mitigate scoring bias caused by some models producing systematically higher or lower logits the scores are normalized across the **model dimension** (I think this might be a typo)?
   1. Combining quation 3 and 4: $y^k_j = \sum_i^N w_i^k \frac{s^k_{i,j}}{\sum_i^N s^k_{i,j}}$, so at the beginning when you have all weights are equal, $y_j^k = \sum_{i=1}^{N} \frac{1}{N} \frac{s^k_{i,j}}{\sum_{i=1}^{N} s^k_{i,j}}=\frac{1}{N}$, meaning that all candidates will have the same score ignoring completely the scores. Is this expected?
   2. If model i produce very high logits, i.e. $s_{i,j} \rightarrow \infty, \forall j$, by equation 3, $s_{i,j} \approx 1$ for i and other r, $s_{r,j}\approx 0$ for all j, then $y_j$ would be dominated by the model i's judgement
2. Why in table 3 SpecFuse improve Chinese FuseEval so much? A deep dive into this might help illustrate why SpecFuse is effective.
3. Slight abuse of notation in equation (3), consider using hat: $\hat{s}^{(k)}_{i,j} \leftarrow ...$
4. Duplicated reference line 378 380
5. Other typos: end of line 4, beginning of line 230, 303

**Ethical Concerns:**

["NO or VERY MINOR ethics concerns only"]

**Final Justification:**

I will maintain my score of 4: Borderline Accept.
The authors' rebuttal clarified key questions about the normalization formula and benchmark results. However, concerns remain.
- The online weighting mechanism, due to its decaying learning rate, has limited ability to adapt to task changes mid-generation.
- Furthermore, the learning rate's design feels heuristic and lacks an ablation study for justification.
- The effectiveness of ensembling "specialist" models (where each model is strong on one sub-task but weak on others) remains a concern.

However, overall I find the core idea of fusing models in the output space—without requiring access to model weights—to be very compelling. With the potential limitations properly discussed, I think the paper will make a meaningful contribution to the field.

**Limitations:**

yes

**Quality:**

3

**Strengths And Weaknesses:**

- Quality
  - Overall solid work with comprehensive experiments on performance and efficiency, comparison with prior methods and ablation studies, clearly outlining the advantages of SpecFuse
  - Although the authors claim that the online feedback mechanism ensures stronger models exert greater influence in the ensemble, this claim is only supported by final performance metrics, without any analysis or visualization of how model weights actually evolve during generation.
  - The learning rate is designed to ensure that the updates become more stable over time (k). However, I am not sure if this would handle situations where there's code switching (First English, then later Chinese; or first English then code at some point). The concern is different models are good at different aspects and the design of weights might not allow it to adapt to changes of tasks during generation.
- Clarity
  - The paper is generally well written, logically organized, and easy to follow.
  - The motivation behind multiplicative weight update is a bit unclear
    - Why are small initial weights a problem at all? Line 175
    - Why is the learning rate designed this way? Equation (7)
  - There are a few typos, formatting or notation issues (see questions/suggestions).
- Significance
  - Interesting idea of combining models at the segment-level with unsupervised online weight adjustment, achieving strong performance with relatively low computational overhead compared to previous ensemble methods.
  - Although the improvement on FuseEval is notable, the SpecFuse (All) for other benchmarks like MMLU, GSM8K (Table 2) is much smaller and can even hurt the model performance (IFEval). It's good that the authors touch on this in the limitation section. However, this slightly contradicts the discussion in line 288-290 "The improvements are more pronounced when stronger models are introduced, **while even weaker models still contribute positively**." Given the fact that one has to host multiple models and sacrifice some inference overhead, it casts some doubt on the practical usefulness of the method.
- Originality
  - Fresh blend of segment-level speculative drafting and verification with online multiplicative-weights voting.

---

> ### Author Rebuttal · Authors · 2025-07-31
>
> Dear Reviewer tSy8,
>
> **Thank you for your thoughtful and thorough review of our manuscript. We truly appreciate the time and effort you dedicated to understanding our work, and your constructive comment. Below, we respond to each of the concerns you raised. We hope this addresses your concerns and helps further improve your evaluation of our work.**
>
> **[Weakness 1: Lack of Visualization of Model Weight Evolution During Generation]**
>
> Thank you for this helpful suggestion, which will help clarify the role of the online feedback mechanism.
>
> * In response, we conducted an additional experiment to analyze how model weights evolve during generation. Specifically, we ensemble three models: Qwen2, GLM-4, and Gemma-2, and track their score-based weights at each decoding step. The task was to "Describe the Forbidden City in Chinese," for which we deliberately selected a setting where one model (Gemma-2) performs relatively poorly. This is because Qwen2 and GLM-4 are pre-trained on large-scale Chinese corpora, while Gemma-2 is less proficient in Chinese.
> The table below shows how the model weights adjust dynamically during the generation process:
>
>   | Round | Qwen2  | GLM-4  | Gemma-2 | Best Candidate Generator |
>   |-------|--------|--------|---------|----------------|
>   |0| 0.3333 |0.3333 |0.3334| |
>   |1| 0.4002 |0.2999 |0.2999|Qwen2|
>   |2| 0.3824 |0.3310 |0.2865|GLM-4|
>   |3| 0.3701 |0.3527 |0.2773|GLM-4|
>   |4| 0.3785 |0.3479 |0.2735|GLM-4 |
>   |5| 0.3708 |0.3612 |0.2680|GLM-4 |
>   |6| 0.3644 |0.3723 |0.2633|GLM-4|
>   |7| 0.3615 |0.3772 |0.2613|GLM-4 |
>   |8| 0.3591 |0.3814 |0.2595|Qwen2 |
>   |9| 0.3606 |0.3830 |0.2564|Qwen2 |
>   |10| 0.3566| 0.3898 |0.2536|GLM-4 |
>   |11| 0.3547| 0.3930 |0.2523|Qwen2 |
>   |12| 0.3575| 0.3913 |0.2512|GLM-4 |
>   |13| 0.3601| 0.3897 |0.2502|Qwen2|
>
>   As shown, the weights of Qwen2 and GLM-4 gradually increase, while the weight of Gemma-2, being the weaker model in this context, decreases over time. This demonstrates that the online feedback mechanism can effectively upweight stronger models and downweight weaker ones as generation progresses.
>
> We sincerely appreciate your valuable question and will include a corresponding line plot of the weight evolution in the final version of the paper to better visualize this dynamic.
>
> **[Weakness2: Adaptation of the online feedback mechanism to task changes during generation.]**
>
> Thank you for your insightful comment. You raise an important scenario where a single user instruction involves multiple tasks, such as switching between languages or modalities.
>
> * First, although the learning rate decays over time, model weights can still adjust meaningfully. This is because stronger-performing models continue to receive higher reward signals during the verification phase, and the final weight updates depend jointly on the reward and the learning rate. Thus, our mechanism still allows for adaptive reweighting, even in later steps.
>
> * To better handle such cases, a promising direction is to dynamically adjust the learning rate based on reward trends, increasing responsiveness when rewards shift consistently. We will continue exploring ways to improve the stability of the SpecFuse ensemble framework in complex situations.
>
> We appreciate your thoughtful feedback and will include this in the paper’s future work section.
>
> **[Weakness 3: Why are small initial weights problematic (Line 175), and what is the rationale behind the learning rate design in Equation (7)?]**
>
> Thank you for your insightful questions. We would like to take this opportunity to clarify the motivation behind our multiplicative weight update strategy.
>
> * **Regarding small initial weights :** Since the number of models N in the ensemble is not fixed, we initialize the model weights uniformly, i.e., each model starts with a weight of 1/N. This means that when N is large, each model begins with a relatively small weight. If we use a constant learning rate across different values of N, the magnitude of weight updates would differ significantly depending on the number of models. For example, with N=2, each model starts with a weight of 0.5, while with N=10, each starts at 0.1. Applying the same learning rate (e.g., 0.01) in both cases would result in disproportionately larger updates when N is small. To mitigate this, our learning rate is designed to scale with the number of models, ensuring that models with small initial weights are not over-updated in larger ensembles.
>
> * **Regarding the design of Equation (7):** The inclusion of N is motivated by the explanation above. The term k is introduced to gradually decay the learning rate over the course of generation. Since the number of generation steps k is typically larger than the number of models N, we apply a square root to k to suppress overly rapid decay and maintain stable updates in later stages.
>
> We hope this clarifies the design choices in our update rule, and we appreciate the reviewer’s thoughtful feedback.
>
> **[Weakness 4: There somewhat contradict the discussion in lines 288–290.]**
>
> Thank you for your thoughtful feedback. We would like to clarify that the results presented in Figure 4 and Table 2 are not contradictory.
>
> * Figure 4 shows that when model performance gaps are small (e.g., 23.31–25.57), even weaker models can complement others and enhance ensemble performance. In contrast, Table 2 illustrates that adding a significantly weaker model may hurt performance. For instance, on IFEval, Qwen, GLM, and Gemma score 41.7, 56.01, and 61.64, respectively. While GLM+Gemma reaches 66.89, adding Qwen drops the ensemble to 62.11.
>
> * In open-domain scenarios, the relative strengths of models are often unknown in advance. As illustrated in Table 1, Gemma performs well in English tasks, whereas Qwen2 is more effective in Chinese. This variation highlights the necessity of integrating multiple models, as doing so enhances the system’s robustness across diverse query types.
>
>   Moreover, recent advances in query-aware model selection [1], which dynamically select a small subset of base models with similar performance for each query, offer a promising way to mitigate this issue. Therefore, model ensembling remains a meaningful and practical approach.
>
> We will revise the relevant statements in the final version to make this distinction clearer.
>
> **[Question1：Why normalize scores across the model dimension)? (I think this might be a typo)]**
>
> Thank you for your careful observation and concern.
>
> * Upon review, you are correct, there is a typographical error in Equation (3): the index should be j=1, not i=1, where j refers to the j-th model that produced the candidate. This correction aligns with our implementation in the provided code (model_verify_in_line function in utils/normal_utils.py).
> We sincerely apologize for the oversight and truly appreciate your careful attention to detail.
>
>   "model dimension" refers to the normalization of the scores that a model produces for all model candidates. Based on the correction of the error in formula 3, the problems of Question1, Question1.1, and Question1.2 will not occur.
>
> * We would like to further clarify with an example:  suppose three models a, b, and c produce scores for the current candidate set as follows: a: [2.3, 2.6, 3.2], b: [1.4, 1.6, 2.2], c: [0.6, 0.3, 0.5].
> Though their score distributions differ in scale, equation (3) normalize each model’s scores over the candidates: a: [0.284, 0.321, 0.395], b: [0.269, 0.308, 0.423] , c: [0.429, 0.214, 0.357].
> By performing this normalization, we ensure that the ensemble decision is not biased toward models that produce higher-magnitude logits.
>
> Thanks for your question again. We hope this clarifies the scoring normalization strategy and resolves your confusion.
>
>
> **[Question2: Why in table 3 SpecFuse improve Chinese FuseEval so much?]**
>
> * Thank you for the question.  We would like to clarify the large improvement of SpecFuse on the Chinese FuseEval benchmark as shown in Table 3. As noted in the table caption, the "Base LLMs" ( Qwen2.5-32B, LLaMA3-70B, Mistral-24B) are compared against Qwen2-72B on FuseEval. Given that Qwen2-72B is particularly strong on Chinese benchmarks, the other base models individually perform worse, leading to win rates below 50% (indicating they lose to Qwen2-72B).
> In contrast, all ensemble methods (MOA, GenFuse, UniTE, SpecFuse) include all four base models, including Qwen2-72B. Thus, the ensembles leverage Qwen2-72B’s strengths, plus complementary knowledge from the other models, allowing SpecFuse to outperform Qwen2-72B (>50% win rate) on FuseEval (Chinese).
>
> * Furthermore, the numbers in parentheses represent the relative improvement of the ensemble model over the best-performing single base model excluding Qwen2-72B. Since the next-best base model performs significantly worse than Qwen2-72B, the relative improvement of SpecFuse appears particularly large.
>
> We appreciate the reviewer’s interest and will consider adding further clarification in the next version.
>
> **[Question3: Slight abuse of notation in equation (3), consider using hat]**
>
> * Thank you for your suggestion. Our initial intent was to keep the use of symbols minimal; however, we are happy to revise it according to your recommendation.
>
> **[Question4: Duplicated reference line 378 380]**
>
> * Thank you for your comment. We will review the references and remove any duplicates accordingly.
>
>
> **[Question5: Other typos: end of line 4, beginning of line 230, 303]**
>
> * Thank you for your careful review. We will go through the paper and correct all typos in the next version.
>
>
> **Thank you for taking the time to read our comments. If we have addressed your concerns, we sincerely hope you can reconsider and improve your evaluation of our work. If you have any other questions, we look forward to further discussions with you. We sincerely await your reply.**
>
> References：
>
> [1] SELECTLLM: Query-Aware Efficient Selection Algorithm for Large Language Models,  ACL 2025

---

> > ### Comment · Reviewer_tSy8 · 2025-08-07
> >
> > Thank you for the detailed response and the additional experiment. While I appreciate the clarifications, several concerns about the method's design and robustness remain.
> >
> > - **W1 & W2 (Online Weighting & Adaptability):** I appreciate the new visualization of the online weight adjustment. However, the table also highlights my original concern: the weight changes become marginal in later steps. This confirms that the learning rate decay in Equation (6) inherently limits the model's ability to adapt to dynamic task changes mid-generation, such as with code-switching.
> > - **W3 (Learning Rate Design):** While the intuition for scaling the learning rate is clear, the specific formulation in Equation (7) feels somewhat heuristic. Since other formulations could also achieve stability, an ablation study comparing different learning rate designs would be necessary to justify this particular choice.
> > - **W4 (Ensembling Weaker Models):** You correctly note that in practice, we don't know which model will perform best on a given task. This raises questions about the robustness of SpecFuse in scenarios with "specialist" models (where each of the N models is strong on one task but weak on all others). In such cases, the ensemble performance could be compromised.
> >
> > Despite these points, I find the core idea of fusing models in the output space—without requiring access to model weights—to be very compelling. This shares a similar insight with a recent paper [1], which uses a similar collaborative decoding approach.
> >
> > A key difference is that SpecFuse collaborates on fixed-length segments, whereas [1] operates at the more dynamic token level. While being already effective, I do think collorating at a fixed-length-segment level is suboptimal both in terms of efficiency and effectiveness. I believe it would be valuable to discuss this distinction and the shared insights with [1] to position this work in a broader context of model collaboration at the output space. Such a discussion would also provide a clearer view of future research directions in this area..
> >
> > I will maintain my positive score and encourage the authors to incorporate these limitations/discussions in the final version.
> >
> > [1] Nudging: Inference-time Alignment of LLMs via Guided Decoding, ACL 2025

---

> > > ### Author Response · Authors · 2025-08-08
> > >
> > > Dear Reviewer tSy8,
> > >
> > > Thank you for taking the time to review our response. We are grateful for your recognition of the core idea of our method. We would like to incorporate these limitations and the related discussion [1] into the final version.
> > >
> > > [1] Nudging: Inference-time Alignment of LLMs via Guided Decoding, ACL 2025
> > >
> > > Best wishes,
> > >
> > > All Authors of the Paper

---

### Official Review · Reviewer_zJQy · 2025-07-03

**Clarity:** 3
**Significance:** 3
**Originality:** 3
**Rating:** 4
**Confidence:** 4

**Summary:**

This paper proposes SpecFuse, a training-free, plug-and-play LLM ensemble framework designed to address several key limitations in current large language model (LLM) ensemble techniques. Rather than relying on token-level aggregation or post-generation fusion, SpecFuse employs a segment-level iterative process that includes drafting, verification and online feedback three stages. It is evaluated on six benchmarks (including FuseEval, MMLU, ARC-C, GSM8K, IFEval, AlpacaEval 2.0) and five model families (7B to 72B), consistently outperforming individual models and strong ensemble baselines (MOA, UniTE, MBR, PairRank, etc.). The framework offers both accuracy gains and latency improvements.

**Questions:**

See weaknesses.

1. Equation 3 normalizes scores across models to mitigate bias (e.g., due to logit scale differences), but this assumes linear comparability. Is this sufficient when combining models with heterogeneous architectures, tokenizers, or calibration characteristics (e.g., mixing a 7B and 70B model)?

**Ethical Concerns:**

["NO or VERY MINOR ethics concerns only"]

**Final Justification:**

Given the rebuttal and other reviews, I have updated scores and reviews accordingly.

**Limitations:**

yes

**Quality:**

3

**Strengths And Weaknesses:**

Strengths:

1. SpecFuse is training-free and plug-and-play. It can directly incorporate diverse LLMs without any task-specific fine-tuning, making it highly accessible for practical use.

2. Strong empirical validation: The method is tested on five major LLM families and six benchmarks, spanning open-domain instruction following, knowledge and reasoning (MMLU, ARC-C, GSM8K, IFEval), and multilingual contexts (English/Chinese). These results show consistent and sometimes large gains over both strong individual models and state-of-the-art ensemble approaches.

Weaknesses:

1. The verify-in-line mechanism requires direct control over attention masks and position IDs, which may not be feasible when using closed-model APIs (e.g., GPT-4, Claude).

2. Most evaluation is conducted via automated metrics, with no human inspection of qualitative failure cases. It remains unclear how SpecFuse handles incoherent, contradictory, or hallucinated candidates, or whether it produces undesirable blends of inconsistent partial segments.

3. As described in Figure 5, the choice of segment length L=10 is reported as optimal based on development set results. However, it remains unclear whether this setting was tuned primarily on short or mid-length inputs. In long-form generation tasks that require extended reasoning or discourse coherence, such a short segment length may lead to fragmented context understanding or premature truncation.

---

> ### Author Rebuttal · Authors · 2025-07-31
>
> Dear Reviewer zJQy,
>
> **Thank you for your thorough review and feedback on our manuscript. We truly appreciate the time and effort you dedicated to understanding our work, and your constructive comments. Below, we respond to each of the concerns you raised. We hope this addresses your concerns and helps further improve your evaluation of our work.**
>
>
> **[Weakness 1：The verify-in-line mechanism may not be feasible when using closed-model APIs (e.g., GPT-4).
> ]**
>
> * Thank you for pointing this out. We acknowledge that the Verify-in-line mechanism requires access to token-level logits and attention masks, which may not be directly accessible when using closed APIs. However, we note that SpecFuse can still be applied to partially closed models like ChatGPT, which expose a limited number of output token-level logits via their APIs. In such cases, we can adopt LLM-as-a-judge strategies [1] to enable compatibility.
>
>
>   Specifically, the core idea of the Verify-in-line method is to obtain normalized token probabilities for all candidates in parallel, thereby improving inference efficiency.
>
>   Suppose we ensemble three models: A (ChatGPT), B and C (open-source models), which produce Candidate 1, Candidate 2, and Candidate 3, respectively. We can construct an input prompt that concatenates the candidates (e.g., [A.Candidate1; B.Candidate2; C.Candidate3]) and instruct ChatGPT to choose the best one from A/B/C. The logits over the choices (A, B, C) provided by ChatGPT's API can then be normalized and used as scores, which are compatible with the Verify-in-line framework and can be combined with scores from open-source models B and C.
>
> We hope this clarification addresses your concerns, and we sincerely appreciate your insightful feedback!
>
>
> **[Weakness 2: How SpecFuse handles incoherent or hallucinated segments]**
>
> * Thank you for your insightful comments regarding the potential risks of incoherence or undesirable blending when combining candidate segments. In our observations, such issues are rare in practice. This is primarily because each segment is selected through an interactive scoring mechanism that considers the context formed by previously chosen segments. The mechanism scores candidates using logits-based values, which is strongly correlated with model perplexity, to evaluate each candidate’s semantic relevance to the preceding text and its overall fluency.
> Intuitively, this means the system favors segments that are more semantically coherent. In other words, if a candidate segment exhibits high perplexity, such as due to semantic contradiction, it is less likely to be selected.
>
> * To further illustrate the robustness of our method, we construct a controlled case. Specifically, we ensemble four candidates: three generated by Qwen2, Mistral, and GLM-4, and one intentionally crafted as a noisy, contradictory input.
>
>   |      | Qwen2  | Mistral  | Glm-4  | Human   Noise | Best   |
>   |-----------------|----------------------------------------------------|------------------------------------------------|---------------------------------------------------|------------------------------------------------------|----------------------------------------------------|
>   | Round   1 Text  | A   person who is hardworking is like a determined | This   person is as dedicated to their work as | A   hard-working person is like a well | Title:   The Cowardly Man| A   person who is hardworking is like a determined |
>   | Round   1 Score | 0.292 | 0.2535 | 0.259  | 0.1955|  |
>   | Round   2 Text  | seed   that refuses to give up until it grows      | river,  carving its path through the | oak,   roots sunk deep in the earth               | slacker   waiting for success without doing anything | river,   carving its path through the    |
>   | Round   2 Score | 0.2609 | 0.312| 0.2646     | 0.1717  |   |
>   | Round   3 Text  | rockiest   of landscapes.  | mountains.   | rocks   and obstacles in its way, never giving up | couch   cushions while watching TV.   rocks   and obstacles in its way, never giving up  |
>   | Round   3 Score | 0.2776     | 0.2388  | 0.312   | 0.1717 |                                                    |
>   | Round   4 Text  | until   it reaches the sea.  | until   it reaches its destination | until its strength fades away.  | …                                                    | until   it reaches its destination    |
>   | Round   4 Score | 0.2749                                             | 0.3765 | 0.2397   | 0.1089   |   |
>
>   As shown in the table above, across all decoding steps, SpecFuse consistently selects segments from high-quality candidates while assigning progressively lower scores to the human-injected noise. This demonstrates that our verification mechanism effectively filters out incoherent segments during the decoding process.
>
> We hope this response addresses your concerns, and we are happy to discuss further if needed. We appreciate the reviewer’s helpful suggestion and will consider including such illustrative examples and options in future releases.
>
> **[Weakness 3: Risk of Fragmented Context Understanding or Premature Truncation]**
>
> * Thank you for raising this important point regarding the impact of segment length on long-form generation. We hope our response to Weakness 2 already addresses part of this concern by clarifying how SpecFuse maintains coherence across segments.
>
> * From a methodological perspective, SpecFuse can be viewed as a greedy algorithm that assembles the most probable output sequence by selecting among candidate segments generated by all ensemble models at each step. In contrast, a standard single-model decoding process constructs an output sequence by only its own produced candidates. As we explained in our earlier response of Weakness 2, previously selected segments are shared as context with all models. During the drafting stage, each model conditions on this shared context to produce its next candidate. In the verification stage, candidates that are semantically inconsistent with the preceding context are unlikely to be selected.
>
> * On the experimental side, our development set for FuseEval includes tasks requiring long-form generation, such as story writing prompts that ask for narratives exceeding 2,000 tokens. We found that a segment length of L=10 performs robustly across both short and long contexts and yields stable improvements over baselines.
>
> *  Furthermore, expanding the search space (e.g., via beam search) could help mitigate the risk of premature truncation. While our current implementation adopts a top-1 selection strategy, we plan to offer beam search as an optional feature in future versions of our code. This would allow users to balance computational cost against the potential for more globally optimal segment combinations.
>
> We hope this clarification addresses your concerns, and we sincerely appreciate your insightful feedback!
>
>
> **[Questions 1：Is this approach sufficient when combining models with heterogeneous architectures, tokenization schemes, or calibration characteristics?]**
>
> * Thank you for your careful inspection.
>
>   First, we would like to acknowledge a typographical error in Equation (3): the index i=1 should actually be j=1, where j refers to the model that generated the j-th candidate. This correction is consistent with our implementation, as shown in the model_verify_in_line function in the provided utils/normal_utils.py script. We apologize for the oversight and appreciate your understanding.
>
> * Building on this correction, we would like to clarify that the normalization in Equation (3) is specifically designed to address potential logit scale disparities across different models. Without normalization, models that systematically produce higher or lower logit values may dominate the final ensemble score, skewing the selection unfairly toward one model. To mitigate this, equation (3) normalize the scores that each model assigns to all candidates before aggregation, effectively removing the impact of differing logit scales. For example, suppose three models a, b, and c produce scores for the current candidate set as follows: a: [2.3, 2.6, 3.2], b: [1.4, 1.6, 2.2], c: [0.6, 0.3, 0.5].
>
>
>   Though their score distributions differ in scale, equation (3) normalize each model’s scores over the candidates: a: [0.284, 0.321, 0.395], b: [0.269, 0.308, 0.423] , c: [0.429, 0.214, 0.357].
>
>   This approach ensures that no model disproportionately influences the final decision due to inherently larger or smaller logits. While this normalization assumes linear comparability, our experiments suggest it is sufficient in practice (24-72b LLM ensembling in Table 3), and is not affected by models with different architectures and tokenizers.
>
> We thank the reviewer again for the valuable feedback, and we will make the clarification more explicit in the final version.
>
>
> **Thank you for taking the time to read our comments. If we have addressed your concerns, we sincerely hope you can reconsider and improve your evaluation of our work. If you have any other questions, we look forward to further discussions with you. We sincerely await your reply.**

---

> > ### Comment · Reviewer_zJQy · 2025-08-06
> > **Thanks for the rebuttal**
> >
> > I would like to thank the authors for the rebuttal, which has somehow tackled weakness #2-#4. Yet stil the unapplicability of the proposed method to closed-source models is its inherent and key limit. Thus I would keep my rating unchanged.

---

> ### Author Response · Authors · 2025-08-08
>
> Dear Reviewer zJQy,
>
> Thank you for taking the time to review our response. We truly appreciate your time and consideration.
>
> Best wishes,
>
> All Authors of the Paper

---

### Note · Authors · 2025-08-13

Dear Chairs and Reviewers,

We would like to express our sincere gratitude to all the chairs and reviewers for the time and effort devoted to evaluating our work. We greatly appreciate the positive scores and encouraging feedback, including remarks that the paper is well-written (tSy8, VDu3, KAZA), that the methodology is innovative (tSy8, VDu3), practical and easy to apply (zJQy), introduces a novel online learning mechanism (tSy8, KAZA), and is supported by strong empirical validation (zJQy, tSy8, VDu3, KAZA).



We are also grateful for the valuable feedback provided, which has offered helpful suggestions for improvement. Based on the feedback, we are committed to making the final revisions of the paper, including incorporating the new experimental results and analysis from the rebuttal, clarifying the applicability of our method to closed-source models and the handling of long-range dependencies, refining theoretical explanations and notation, expanding discussions with the latest related work, and strengthening the discussion on limitations and potential societal impacts.

Once again, thank you for your consideration and valuable time.

Warm regards,

All Authors of the Paper

---

### Decision · Program_Chairs · 2025-09-17

**Decision:**

Accept (poster)

**Comment:**

This paper introduces SpecFuse, a training-free plug-and-play framework for ensembling LLMs. The method works by iteratively drafting candidate text segments from multiple LLMs and then scoring all candidates across all models and selecting (and broadcasting to all models) the segment with the highest combined score. An online feedback mechanism dynamically adjusts the influence of each model based on its performance, allowing stronger models to contribute more. This segment-level collaboration improves performance and efficiency compared to prior ensemble methods.

**Strengths:**
*   The method is novel, taking inspiration from speculative decoding, but applying these ideas in the model ensembling domain. Unlike Speculative decoding, there is no lossless speedup guarantee here, but the gains are by empirically getting better results via ensembling, without having to wait for all models to complete their full generations before fusing them.
* The online feedback mechanism is also a nice addition and provides additional gains.
*   The paper provides strong and comprehensive empirical validation, demonstrating consistent improvements over strong baselines across various tasks and model sizes. The authors provided additional experiments during the rebuttal that further strengthened these claims.

**Weaknesses:**
*   The method relies on logit access, making it difficult to apply to closed-source models. The authors propose to use LLM-as-judge approach with API calls to get scores, but that will probably make the whole process too slow and expensive.
*   Reviewers raised questions about the online weighting mechanism, such as its robustness to mid-generation task shifts and the justification for its specific design (tSy8, VDu3).
* Some terminology can be more precise to avoid confusions, like the use of “fusion” instead of selection, and clarifying that there is no speculative decoding guarantees.